# Functionality of chimeric TssA proteins in the type VI secretion system reveals sheath docking specificity within their N-terminal domains

Selina Fecht[1], Patricia Paracuellos[1], Sujatha Subramoni[2], Casandra Ai Zhu Tan [2], Aravindan Ilangovan [3], Tiago R. D. Costa [1] & Alain Filloux [1,2] ✉

The genome of *Pseudomonas aeruginosa* encodes three type VI secretion systems, each comprising a dozen distinct proteins, which deliver toxins upon T6SS sheath contraction. The least conserved T6SS component, TssA, has variations in size which influence domain organisation and structure. Here we show that the TssA Nt1 domain interacts directly with the sheath in a specific manner, while the C-terminus is essential for oligomerisation. We built chimeric TssA proteins by swapping C-termini and showed that these can be functional even when made of domains from different TssA sub-groups. Functional specificity requires the Nt1 domain, while the origin of the C-terminal domain is more permissive for T6SS function. We identify two regions in short TssA proteins, loop and hairpin, that contribute to sheath binding. We propose a docking mechanism of TssA proteins with the sheath, and a model for how sheath assembly is coordinated by TssA proteins from this position.

Bacteria are the most abundant living organisms on earth and have been competing with each other, and with other microorganisms, for billions of years[1,2]. The myriad of antagonistic strategies they have developed, include the use of supramolecular devices, like the type VI secretion system (T6SS), which provides a clear advantage in contact-dependent combat[3,4]. The T6SS is a contractile nanomachine that transports effector proteins out of the cell, including into neighbouring cells[5]. Effector proteins, such as anti-bacterial toxins, can be loaded on the VgrG-PAAR puncturing device or in Hcp rings[6–11], which stack to form a spear that is wrapped in a contractile sheath, TssBC[12–14]. This tail complex extends into the cytoplasm from a baseplate (TssEFGK) which is docked onto a membrane complex (TssJLM)[15–17]. Once the sheath reaches the opposite side of the cell it contracts, propelling the toxin-loaded spear into prey cells[18,19], and resulting in prey killing or growth arrest[20].

*Pseudomonas aeruginosa* is a ubiquitous, opportunistic Gram-negative bacterial pathogen, responsible for acute and chronic infections[21,22], including in the lungs and airways of cystic fibrosis patients[23]. Remarkably, there are three T6SS clusters within the *P. aeruginosa* genome (H1-, H2- and H3-T6SS)[24,25], each encoding all the core structural components required to assemble a functional T6SS (*tssA-M*). The presence of multiple T6SS machineries is hypothesised to be linked with variations In expression and functional/mechanistic properties. While there are some conserved pathways regulating the expression of these three machineries, distinct pathways drive variation in their activity[26–31]. For example, the H1-T6SS is post-translationally regulated by the threonine phosphorylation pathway[32,33], which is associated with the Tit-for-Tat mechanism[34] to fire in response to detecting contact from an aggressive opponent[35]. We have previously identified variation in the abundance, position and

¹CBRB Centre for Bacterial Resistance Biology, Department of Life Sciences, Imperial College London, London SW7 2AZ, UK. ²Singapore Centre for Environmental Life Sciences Engineering, Nanyang Technological University, Singapore 637551, Singapore. ³School of Biological and Behavioural Sciences, Queen Mary University of London, London E1 4NS, UK. ✉e-mail: a.filloux@ntu.edu.sg

firing dynamics between these three T6SSs[36]. The H1-T6SS contracts very rapidly after assembling, while H2-T6SS sheath structures are maintained in the extended conformation for a prolonged period before firing[36]. These varying firing dynamics have been linked to variations in the TssA protein and the interaction it has with accessory components such as TagA or TagB[18,36,37].

In this study we have used *P. aeruginosa* as a model to investigate the diversity and specificity of T6SSs and their cognate TssA proteins. This is due to the great variability in the class of proteins encoded by *tssA* genes between different T6SSs: varying in size, domain organisation, symmetry, and structure[19,36–40]. One way this variation has been classified is grouping TssA proteins by size, i.e. long (TssA$_L$: 470–540 aa or 55–60 kDa) and short (TssA$_S$: 340–380 aa or ~40 kDa)[36]. *P. aeruginosa* encodes three TssA proteins: the short TssA1 and TssA3, and the long TssA2. This grouping of TssA proteins corresponds with differences in domain organisation, with specific discrete domains connected by flexible linkers associated with long and short TssA proteins[36,37]. All TssA proteins share a N-terminal Nt1 domain containing the characteristic ImpA_N region[38]. In TssA$_S$ proteins this is linked to a C-terminal domain, while in TssA$_L$ proteins this is linked to a Nt2 domain connected to a distinct C-terminal domain[19,38] (Fig. 1 and Supplementary Fig. 1). This distinct domain organisation contributes to structural diversity, where despite their differences, the C-terminal domains of both long and short TssA proteins oligomerise into ring-shaped structures, but the dimensions and symmetries of these rings vary[19,36–39] (Fig. 1). The C-terminal domain ring of the TssA$_S$ of *Burkholderia cenocepacia* (TssA$^{BC}$) is made of 32 monomers, forming a large ring of 16-fold symmetry with an external diameter of 200 Å and a lumen of 100 Å[38]. In contrast, the C-terminal domain rings of TssA$_L$ proteins are formed of 10–12 monomers, forming a narrow ring of five- or six-fold symmetry with external diameters of ~100 Å and lumens of ~25–50 Å[19,37]. The TssA$_L$ C-terminal domain ring is surrounded by Nt2 domain dimers, and for both long and short TssA proteins Nt1 domains are anticipated to be flexibly associated with the periphery[38].

Despite their structural diversity, both long and short TssA proteins interact with components of the membrane complex, baseplate and tail complex, which has sparked debate about the functional role of TssA within the T6SS[19,37–41]. Both long and short TssA proteins are important, in some cases essential, for sheath assembly, and have been demonstrated to interact with sheath components[19,37–39]. For TssA$_L$ proteins, sheath interaction has been suggested to involve the Nt2 domain, which is absent in TssA$_S$ proteins[19,37]. One suggestion has thus been that the nature of these sheath interactions vary: with TssA$_L$ proteins positioned at the distal end of the sheath[19] and TssA$_S$ proteins localised at the proximal end, as a stationary component of the baseplate[37,39]. However, subsequent studies have identified that TssA proteins of both classes localise to the membrane-baseplate complex, where they are proposed to be key for initiating the assembly of the sheath and are then displaced to the distal end of the growing sheath, where they are predicted to coordinate sheath polymerisation[19,36,37,42].

It remains to be fully explained how similarities in TssA function and positioning within the T6SS are achieved by these diverse structures.

Using the full set of *P. aeruginosa* TssA proteins we showed that these proteins function specifically with their cognate T6SS and this specificity can be linked to the direct interaction with the associated TssBC sheath complex via their Nt1 domain. This specific interaction drives the TssA component to its cognate sheath, and remarkably, this targeting can be altered using chimeric proteins in which heterologous N- and C-termini of different TssA proteins are exchanged. We showed that the main role of the C-terminus is in TssA ring formation and that, despite structural differences, functionality is preserved even in chimeras comprising domains originating from a mixture of short and long TssA proteins. One further key observation in our study is the demonstration that two structural features of the TssA Nt1 domain varied when comparing the homology molecular models of two short TssA proteins, TssA1 and TssA3: a loop and a hairpin. Exchange of these secondary structures between the two TssA$_S$ proteins resulted in loss of interaction with the cognate TssBC. We could model and dock these TssA structures into the three-dimensional TssBC structure and suggest an original mechanism for assembly during sheath elongation.

## Results

### TssA proteins have functional specificity for their cognate T6SS

Given the presence of multiple *tssA* genes in the *P. aeruginosa* genome, we were curious whether the encoded proteins could be functional in a heterologous context. The activity of each T6SS was therefore assessed in the presence of each TssA in an appropriate T6SS background, using several approaches including secretion of Hcp[7,43], bacterial competition and killing assay[44] or assembly of T6SS sheath structures[36,45]. The activity of the H3-T6SS could only be assessed by quantifying assembly of sheath structures, as sheath contraction could not be stimulated, preventing assessment by secretion or prey killing.

The functionality of the H1-T6SS was impaired by the deletion of either the *tssB1* gene encoding the sheath component or the cognate *tssA1* gene. We then complemented the *tssA1* mutant with each of the three *tssA* genes. The activity of the H1-T6SS secretion system was exclusively dependent on the presence of *tssA1*, with Hcp1 secretion or prey killing only possible with the *tssA1* mutant complemented by *tssA1* expressed from a plasmid in trans, while introduction of neither *tssA2* nor *tssA3* could restore these H1-T6SS phenotypes (Fig. 2a, b, Supplementary Fig. 2a). Similarly, for both the H2- and H3-T6SSs, activity was only restored upon introduction of the cognate *tssA* (Fig. 2c–e, Supplementary Fig. 2b), identifying a functional specificity of TssA proteins relative to their cognate T6SS.

### Specificity of interaction between TssA and the TssBC sheath lies in the TssA Nt1 domain

To investigate the observed functional specificity, the positioning of TssA proteins within the T6SS was considered. Despite their differences, both long and short TssA proteins have been observed to

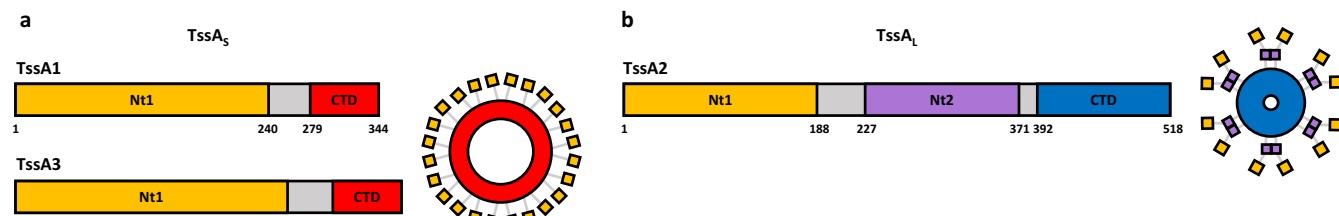

**Fig. 1 | TssA proteins form ring structures with peripheral conserved Nt1 domains. a** In TssA$_S$ proteins, such as *P. aeruginosa* TssA1 and TssA3, a flexible linker (grey) connects the Nt1 domain (yellow) to the C-terminal domain (CTD) (red). Oligomerisation of the TssA$_S$ CTD forms a ring-shaped structure with peripheral Nt1 domains. **b** In TssA$_L$ proteins, such as *P. aeruginosa* TssA2, a flexible linker connects the Nt1 domain to the Nt2 domain (purple). The Nt2 domain is then connected by a flexible linker to a distinct TssA$_L$ CTD (blue). Oligomerisation of the TssA$_L$ CTD forms a ring-shaped structure with peripheral Nt2 domain dimers connecting to Nt1 domains.

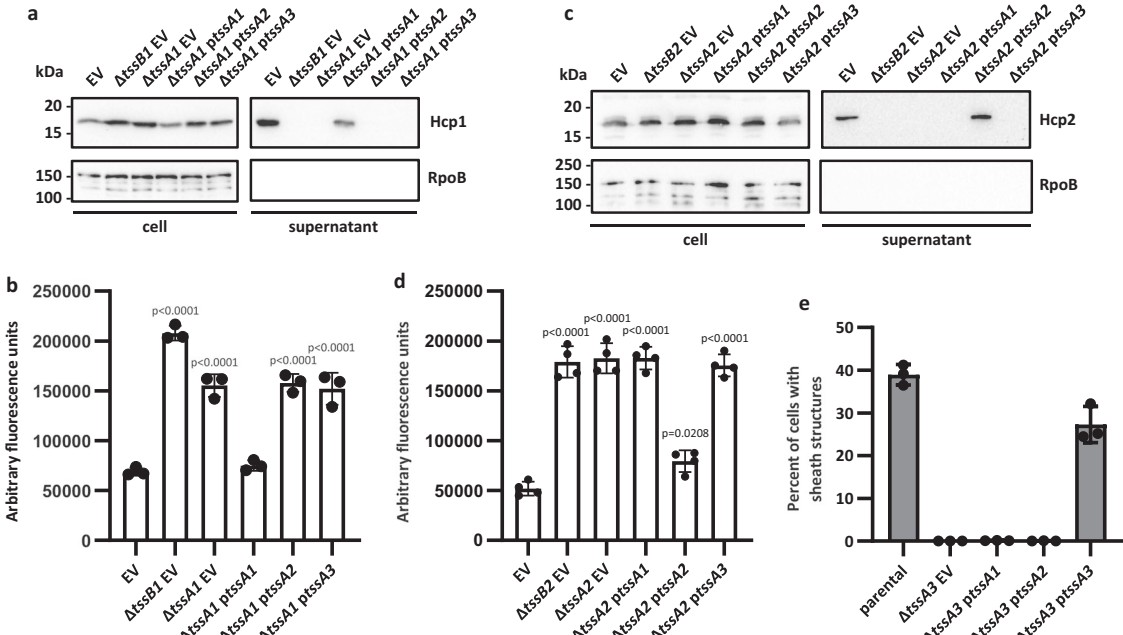

**Fig. 2 | Functional specificity of TssA proteins for their cognate T6SS.** To assess complementation of a *tssA* deletion, the pBBR1MCS4 vector was introduced either as empty vector (EV) or encoding *tssA1*, *tssA2* or *tssA3* (p*tssA1/2/3*). **a** Western blot of a H1-T6SS secretion assay in the PAO1Δ*rsmA*Δ*rsmN* background to assess the levels of the secretion marker Hcp1 in the supernatant sample. RpoB was used as a bacterial lysis control. Representative of 3 repeats. **b** H1-T6SS competition assay to assess the T6SS-dependent killing of a GFP-encoding *E. coli* prey by *P. aeruginosa* attacker strains in the PAO1Δ*rsmA*Δ*rsmN* background. (*n* = 3 independent experiments). **c** Western blot of a H2-T6SS secretion assay in the PAO1Δ*rsmA* background to assess the levels of the secretion marker Hcp2 in the supernatant sample. RpoB was used as a bacterial lysis control. Representative of 3 repeats. **d** H2-T6SS competition assay to assess the T6SS-dependent killing of a GFP-encoding *E. coli* prey by *P. aeruginosa* attacker strains in the PAO1Δ*rsmA* background. (*n* = 4 independent

experiments). **e** Quantification of the percentage of cells with TssB3-mScarlet sheath structures by fluorescence microscopy in the PAO1Δ*rsmA tssB3-mScarlet-I* background. In the parental strain, sheath structures were formed in 38.9% of cells (*n* = 10582 cells examined), upon deletion of *tssA3* with the EV present this fell to 0.036% (*n* = 16822 cells examined), upon introduction of *tssA1* 0.14% of cells formed sheath structures (*n* = 11205 cells examined) and with *tssA2* this was 0.082% (*n* = 8500 cells examined). Upon introduction of *tssA3* 27.2% of cells formed sheath structures (*n* = 16253 cells examined). Data were taken from two fields of view for three experiments. Statistical testing was conducted by one-way ANOVA with Dunnett's multiple comparisons test, each strain was compared to parental EV strains. Values are presented as means, error bars represent standard deviation. Source data are provided as a source data file.

interact with components of each of the T6SS sub-complexes: membrane complex, baseplate complex and tail complex[19,37–39,41]. This includes interactions with sheath proteins, where the positioning of TssA complexes at the distal end of the polymerising sheath has been demonstrated by fluorescence microscopy[19,36–39,42]. Here we re-investigate the interaction with the TssBC sheath as a specificity-driven constraint during T6SS assembly.

Firstly, a bacterial two-hybrid (BTH) approach[46] was used to investigate the interactions between each TssA and their cognate TssBC sheath. Each *tssA* gene and the gene encoding the cognate sheath components were cloned into complementary BTH vectors[39,47,48]. Using this approach, none of the TssA proteins were able to significantly interact with either TssB or TssC independently (Fig. 3a), although interactions of TssA proteins with individual sheath components have been demonstrated previously[19,37–39,41]. Instead, only when both *tssB* and *tssC* were cloned into the BTH vector was clear interaction between TssA and the cognate sheath readily detected (Fig. 3a).

As in a previous report an N-terminal (Nt1-Nt2) region of the *Escherichia coli* TssA[EC] was proposed to interact with the T6SS sheath[19], and considering that there are no Nt2 domains in TssA1 or TssA3, we investigated whether the Nt1 domain could be involved in sheath interaction. A truncation comprising the Nt1 domain of each of TssA1, TssA2 and TssA3 was therefore tested for interaction with the cognate sheath components. Not only did each Nt1 domain interact with the cognate sheath, but this appeared to be extremely specific since no cross-interaction was observed in heterologous combinations (Fig. 3b). Furthermore, TssA C-terminal regions, including that of

TssA2 containing the Nt2 domain, did not interact with sheath components (Supplementary Fig. 3a).

Interaction with another widely reported TssA interaction partner, the baseplate component TssK[19,36–39,41], was also assessed using each of the *P. aeruginosa* TssA proteins. Each TssA interacted with its cognate TssK protein (Supplementary Fig. 3b), however the contribution of individual domains could not be elucidated at this stage.

## TssA C-terminus is involved in multimerization and ring formation

A remarkable feature of TssA proteins is the ability to form multimers, which occurs through the C-terminal domain[38,39]. Using the BTH approach and appropriate cloning of DNA regions corresponding to each TssA or TssA domain, we could demonstrate not only that oligomerisation occurred (Supplementary Fig. 3c), but it exclusively involved C-terminal regions and was also highly specific as there was no cross-interaction between heterologous C-terminal regions (Supplementary Fig. 3d).

We subsequently purified the C-terminal domains of TssA1 and TssA3, which both belong to the TssA_S family. The corresponding regions were cloned in pETDuet-1 and subsequently purified by affinity chromatography and size exclusion chromatography (Fig. 4a, b and Supplementary Fig. 4). The highly pure products were analysed by negative stain electron microscopy (EM), and both showed ring-shaped structures with a large internal lumen (Fig. 4c, d). The TssA1 C-terminal domain was further analysed using cryo-EM from which single particles were picked and used to generate 2D class averages (Fig. 4e). From these 2D class averages the TssA1 C-terminal domain

was identified to form a very defined ring structure with a ~100 Å internal diameter and a ~170 Å external diameter, consistent with previous reports[39]. Unfortunately, given the low number of particles in our dataset it was difficult to estimate the symmetry of the ring from the 2D class averages. Further high-resolution data will be required to unveil the symmetry of the complex. Yet, either six- or 12-fold sym-

metry has previously been suggested[39], where a 12-fold symmetry double dodecamer would align more closely with the reported 16-fold symmetry TssA[BC] oligomer[38]. As for TssA3, single particles were picked from the negative stain EM dataset to generate 2D class averages (Fig. 4f). Interestingly, these TssA3 C-terminal domain rings display some flexibility and are less defined in their dimensions when compared to the TssA1 C-terminal domain. The relative flexibility of the TssA3 ring may be a factor contributing to the difference in elution volumes of these two C-terminal domains despite their very similar size (Supplementary Fig. 4). Notably, based on the visual assessment of the 2D class averages we can estimate that the complex of the TssA3 C-terminal domain could present a 12-fold symmetry arrangement.

We also attempted to purify the C-terminal domain of TssA2, a member of the TssA$_L$ family. However, a sample could not be obtained that was sufficiently stable for EM analysis (Supplementary Fig. 5). This likely reflects a difference in biochemical properties and/or different structural organisation of the TssA2 C-terminus as compared to those of the TssA$_S$ proteins, as was reported with the long TssA[EC], whose ring structure is drastically different including a much narrower lumen[19].

### A cognate Nt1 domain suffices to yield functional chimeric TssA proteins in the T6SS

To better characterise the contribution of the TssA domains to T6SS activity, 8 chimeric TssA proteins were engineered with mixed domains originating from distinct TssA proteins (Fig. 5). The Nt1 domain of each TssA was combined with the C-terminal regions of each of the other two TssA proteins. In the case of TssA2, both the C-terminal domain alone and in combination with the Nt2 domain were combined with TssA1 and TssA3 Nt1 domains. The production and stability of these chimeric TssA proteins was assessed through the addition of C-terminal hemagglutinin (HA) tags and western blot analysis (Supplementary Fig. 6). Subsequently, we assessed the activity

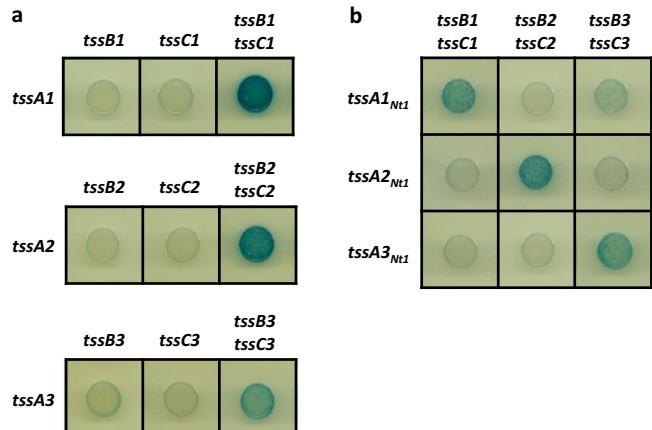

**Fig. 3 | Interactions between TssA proteins and their contractile sheath components.** Protein-protein interactions were assessed by bacterial two-hybrid (BTH), where bacteria appear blue upon interaction of fusion proteins, and white in the absence of interaction. **a** BTH interactions between *P. aeruginosa* TssA proteins and their respective sheath components, with TssB and TssC components produced alone and in combination. **b** BTH interactions between each TssA Nt1 domain with each sheath complex.

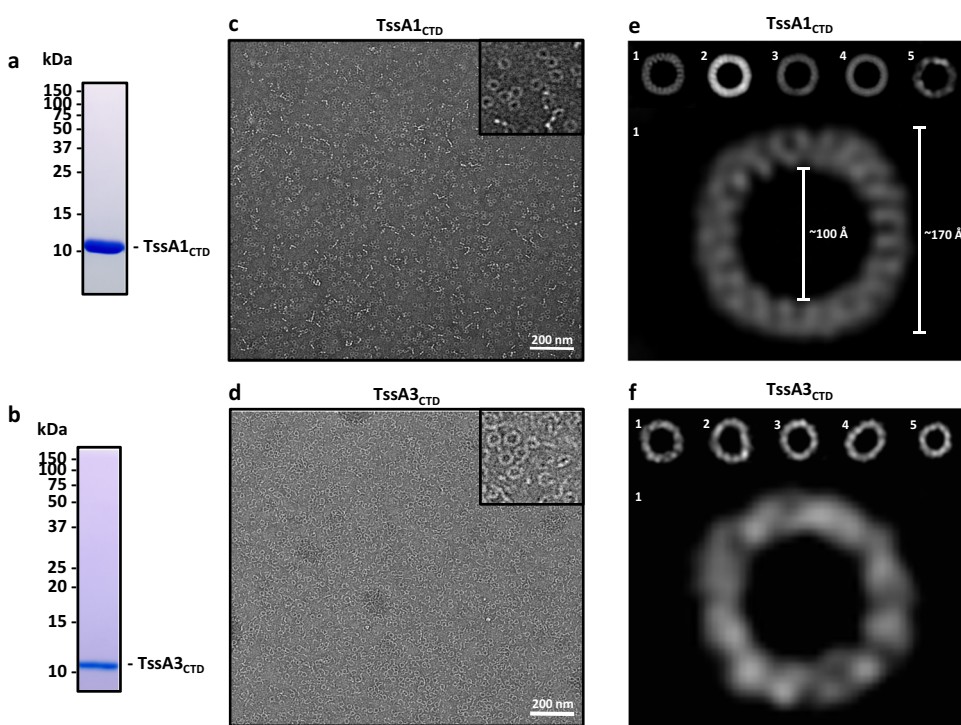

**Fig. 4 | Structural analysis of TssA$_S$ C-terminal domains (CTDs). a** SDS-PAGE of the purified TssA1 CTD, sample taken from the middle of the elution peak (Supplementary Fig. 4a). **b** SDS-PAGE of the purified TssA3 CTD, sample taken from the middle of the elution peak (Supplementary Fig. 4b). **c** Representative negative stain EM images of TssA1 CTD single particles. Scale bar represents 200 nm. **d** Representative negative stain EM images of TssA3 CTD single particles.

**e** Representative 2D class averages of TssA1 CTD particles picked from cryo-EM images. Class average 1 is enlarged (below) with ring dimensions shown. **f** Representative 2D class averages of TssA3 CTD particles picked from negative stain EM images. Class average 1 is enlarged (below). Scale is not comparable between **e** and **f**. Source data are provided as a source data file.

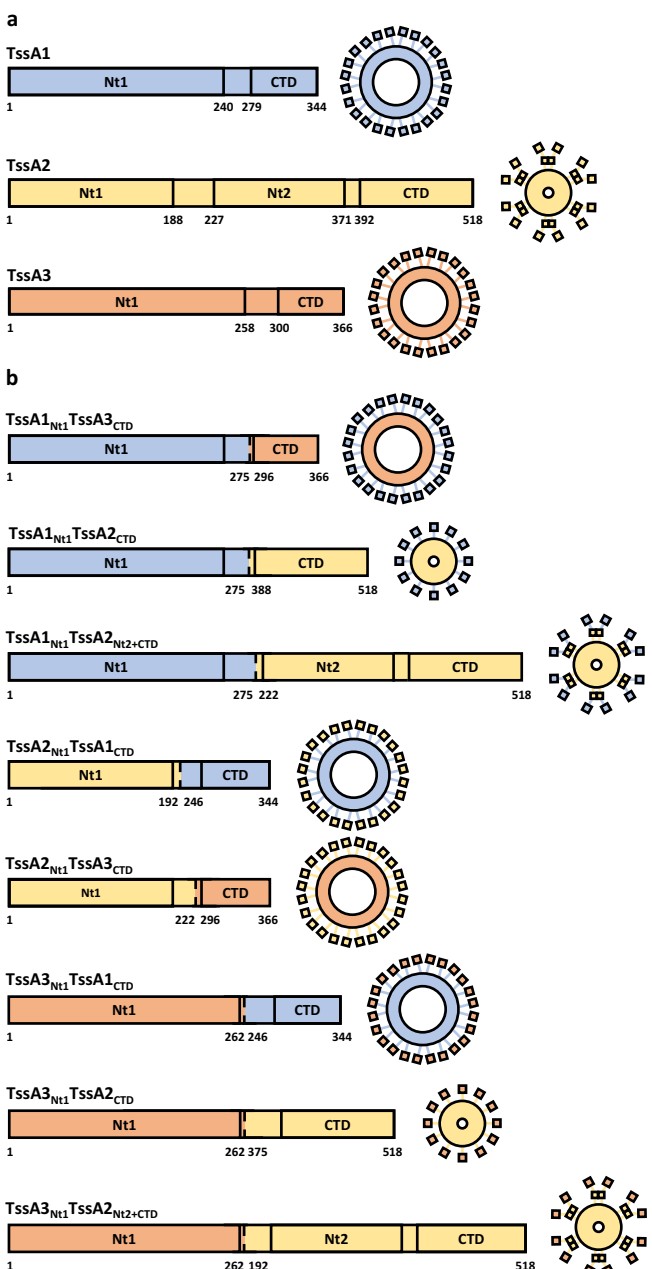

**Fig. 5 | Schematic of wildtype and chimeric TssA proteins.** Residues taken from each component protein and predicted structures of chimeric TssA protein complexes are given. **a** Schematic of wildtype *P. aeruginosa* TssA proteins with domain positions and predicted structural organisation. **b** Schematic of chimeric TssA proteins. The Nt1, Nt2 and C-terminal domains (CTDs) of the TssA1 (blue), TssA2 (yellow) and TssA3 (orange) were exchanged to generate eight chimeric TssA proteins with mixed domains.

of the chimeras in each of the T6SSs using the previously described functional assays.

When considering the functionality of the H1-T6SS, we complemented a *tssA1* mutant with the wild-type (WT) allele and TssA1 domain-containing chimeras. When using chimeras encoding a non-cognate Nt1 domain (i.e. TssA2 or TssA3 Nt1 domains), no complementation could be observed in any of the assays (Fig. 6a, b). However, with a chimera containing the cognate Nt1 domain fused to the non-cognate C-terminal domain of TssA3 (TssA1$_{Nt1}$-TssA3$_{CTD}$) complementation was very effective and at a comparable level to WT TssA1 (Fig. 6a–c and Supplementary Fig. 7a). Instead, when the chimera included the C-terminus of TssA2 no complementation could be

observed (Fig. 6a, b), which likely reflects the very different structural characteristics of these C-terminal regions in long or short TssA proteins (although TssA1$_{Nt1}$-TssA2$_{CTD}$ was also present at lower levels than other chimeras in expression tests (Supplementary Fig. 6). This applies also for the H3-T6SS, where the key factor in the chimera was the presence of the TssA3 Nt1 domain, which, when fused to the C-terminal domain of TssA1 (TssA3$_{Nt1}$-TssA1$_{CTD}$) was able to complement a *tssA3* mutant as effectively as the full length *tssA3* when analysing the abundance of H3-T6SS sheath structures (Fig. 6d and Supplementary Fig. 7b). Again, a chimera including the C-terminus of the long TssA2 with the N-terminus of the short TssA3 could not rescue functionality, despite being well expressed (Fig. 6d and Supplementary Fig. 6).

Given that chimeras formed by switching C-terminal domains between *P. aeruginosa* TssA$_S$ proteins were functionally indistinguishable from WT, cross-species chimeric proteins were generated exchanging the C-terminal domains of the short TssA1 of *P. aeruginosa* and the TssA1 of *P. putida* (Supplementary Fig. 7)[49]. As with the previously described exchange of C-terminal domains between TssA$_S$ proteins, cross-species chimeras containing the cognate Nt1 domain were fully functional in the H1-T6SS (Fig. 6e, f).

Importantly, we demonstrated that it is not systematic that a functional chimera cannot be built between mixed domains of long and short TssA proteins. Indeed, when using a chimera containing the long TssA2 Nt1 domain and trying to complement a *tssA2* mutant, grafting the C-terminal domains of either short TssA1 (TssA2$_{Nt1}$-TssA1$_{CTD}$) or TssA3 (TssA2$_{Nt1}$-TssA3$_{CTD}$) yielded a functional protein for secretion, bacterial competition, or sheath assembly (Fig. 7 and Supplementary Fig. 8c). Equivalent H2-T6SS activity was achieved with TssA2 Nt1 and Nt2 domains, connected to the C-terminal domains of the short TssAs (TssA2$_{Nt1+Nt2}$-TssA1$_{CTD}$ or TssA2$_{Nt1+Nt2}$-TssA3$_{CTD}$) (Supplementary Fig. 9). We thus concluded that the key element for functionality is the recognition by TssA of the core machine, and this occurs through the Nt1 domain, whereas the ring formed by the C-terminal domain is more permissive and can be accommodated in some exchanges. Yet, a chimera between the N-terminal regions from a long TssA2 with the C-terminal domain of a short TssA is slightly impaired in function, as could be seen by the reduced level of Hcp secretion (Fig. 7a) or bacterial competition (Fig. 7b and Supplementary Fig. 9b), which is not the case when examining chimeras between TssA proteins of the same group (Fig. 6a, b, e, f).

## Specific loop and hairpin secondary structures identified within the Nt1 domain

Our work demonstrates a tight specificity between the Nt1 domain and the cognate sheath, which is instrumental to T6SS function, in contrast to the permissive nature of the C-terminal domain. This Nt1 domain specificity is observed even when considering TssA proteins belonging to the same group, such as the short TssA1 and TssA3, which are anticipated to be more similar compared to that of a long TssA, as demonstrated by homology molecular modelling of each of the *P. aeruginosa* TssA Nt1 domains (Supplementary Fig. 10). Here we further investigated this specificity through direct comparison between TssA1 and TssA3 Nt1 domains.

We hypothesised that specificity may be linked to non-conserved residues exposed on the surface. To identify non-conserved residues, Nt1 domain sequences were aligned using Clustal Omega[50] (Supplementary Fig. 11a). Using the homology molecular model of the TssA1 Nt1 domain, the solvent accessible surface area was estimated using GetArea with residues assigned a value of over 80% considered to be solvent accessible[51] (Supplementary Fig. 11b). By cross-referencing these two datasets, 16 residues were identified that were both non-conserved between TssA1 and TssA3 and predicted to be surface exposed in the TssA1 Nt1 domain (D14, D19, M42, D44, D113, D116, R146, F167, S169, E170, D185, D187, R194, S216, Q218 and L238) (Supplementary Fig. 11c). An initial seven of these residues were substituted

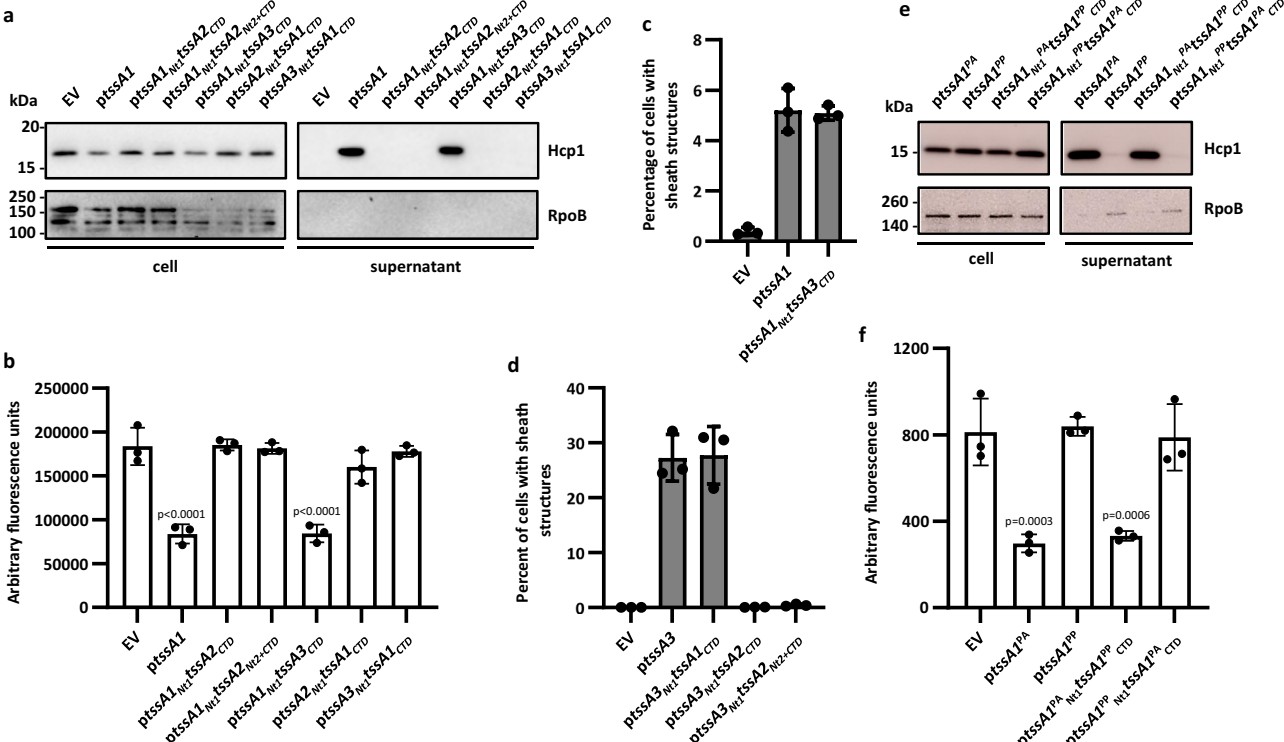

**Fig. 6 | Activity of H1- and H3-T6SSs with chimeric TssA proteins.** To assess complementation of the *tssA* deletions, the pBBR1MCS4 vector was introduced either as EV, or encoding *tssA1* (p*tssA1*), *tssA3* (p*tssA3*) or chimeric *tssA*s. **a** Western blot of a H1-T6SS secretion assay in the PAO1Δ*rsmA*Δ*rsmN*Δ*tssA1* background, to assess the levels of secretion marker Hcp1 in the supernatant sample. RpoB was used as a bacterial lysis control. Representative of 3 repeats. **b** H1-T6SS competition assay to assess the T6SS-dependent killing of a GFP-encoding *E. coli* prey by *P. aeruginosa* attacker strains in the PAO1Δ*rsmA*Δ*rsmN*Δ*tssA1* background. ($n = 3$ independent experiments). **c** Quantification of the percentage of cells with TssB1-mScarlet sheath structures by fluorescence microscopy in the PAO1Δ*retS*Δ*tssA1* *tssB1-mScarlet-I* background. The percentage of cells with sheath structures was 0.37% ($n = 10017$ cells examined) with *tssA1* deletion, upon introduction of *tssA1* or *tssA1_{Nt1}tssA3_{CTD}* on a plasmid this rose to 4.95% ($n = 8117$ cells examined) and 5.26% ($n = 1055$ cells examined 3) respectively. Data was taken from two fields of view for three experiments. **d** Quantification of the percentage of cells with TssB3-mScarlet sheath structures by fluorescence microscopy in the PAO1Δ*rsmA*Δ*tssA3* *tssB3-mScarlet-I* background. Sheath structures were formed in 0.04% ($n = 16822$ cells

examined) of cells in the absence of *tssA3*, this rose to 27.23% ($n = 16253$ cells examined) with expression of *tssA3* on a plasmid and 28.07% ($n = 10456$ cells examined) with *tssA3_{Nt1}tssA1_{CTD}* on a plasmid. With expression of *tssA3_{Nt1}tssA2_{CTD}* or *tssA3_{Nt1}tssA2_{Nt2+CTD}* from a plasmid, 0.08% ($n = 8751$ cells examined) and 0.41% ($n = 9472$ cells examined) of cells formed sheath structures respectively. Data was taken from two fields of view for three experiments. **e** Western blot of a H1-T6SS secretion assay with cross-species chimeras, with domains switched between *P. aeruginosa* TssA1 (TssA1^PA) and *P. putida* TssA1 (TssA1^PP), in the PAO1Δ*rsmA*Δ*rsmN*Δ*tssA1* background, to assess the levels of secretion marker Hcp1 in the supernatant sample. RpoB was used as a bacterial lysis control. Representative of 3 repeats. **f** H1-T6SS competition assay to assess the T6SS-dependent killing of a GFP-encoding *E. coli* prey by *P. aeruginosa* attacker strains expressing *P. putida* wildtype and cross-species chimeric TssA proteins in the PAO1Δ*rsmA*Δ*rsmN*Δ*tssA1* background. ($n = 3$ independent experiments). Statistical testing was conducted by one-way ANOVA with Dunnett's multiple comparisons test, each strain was compared to the parental *tssA1* EV strain. Values are presented as means, error bars represent standard deviation. Source data are provided as a source data file.

into alanine in TssA1, and interactions with WT TssA1 and the H1-T6SS sheath were assessed by BTH (Supplementary Fig. 11d). Each of these TssA1 variants was able to interact with both WT TssA1 and the sheath components. This suggests that these mutations did not disrupt the overall fold of the protein, but also that none of these individual residues were essential for interaction with the sheath and thus H1-T6SS function. It was thus proposed that multiple residues might need to be altered to disrupt the interaction.

Next, using the Dali structural alignment server[52], we superimposed the predicted Nt1 domain models for TssA1 and TssA3. Two regions were identified as variable between these Nt1 domains, that we named the hairpin and the loop (Fig. 8a and Supplementary Fig. 12). The loop shares the same number of residues between TssA1 (Q37-E53) and TssA3 (D46-D62), but conservation of residues is low (12% identity). Differences are more drastic in the hairpin (TssA1 F169-L172, TssA3 A174-Q191), with an 11 aa insertion present in TssA3. The hairpin is located in an extension to the Nt1 domain structure found only in TssA_S proteins, similarly the loop sequence is predicted to be absent in TssA_L proteins[38]. We hypothesised that these differences in hairpin/loop regions could account for the

specificity in sheath interaction in short TssA proteins. To assess this, the hairpin and loop sequences of TssA1 were exchanged for those of TssA3 and vice et versa. These engineered TssA1 and TssA3 proteins were then tested for their ability to work in a H1- or H3-T6SS context, respectively. The production and stability of these proteins was assessed using HA tagged versions (TssA1-TssA3_{hairpin+loop} was not stable, so was excluded) (Supplementary Fig. 13a). When using BTH to assess the self-interaction ability of these different TssA variants, all yielded a significant signal, however, when assessing interaction with their cognate sheath, all signals were impaired or lost, implicating these secondary structures in the interaction with the T6SS sheath (Fig. 8b, c). When assessing H3-T6SS functionality and sheath assembly, swapping either the loop or hairpin was readily deleterious to the function (Fig. 8d and Supplementary Fig. 13b). Yet in case of the H1-T6SS function, for Hcp1 secretion (Fig. 8e) or sheath assembly (Fig. 8f and Supplementary Fig. 13c), the most drastic effect was observed upon loop replacement (TssA1-TssA3_{loop}). Overall, we thus concluded that there are structural elements in the TssA_S Nt1 domain which may be essential for driving proper and specific interaction between TssA and the T6SS sheath.

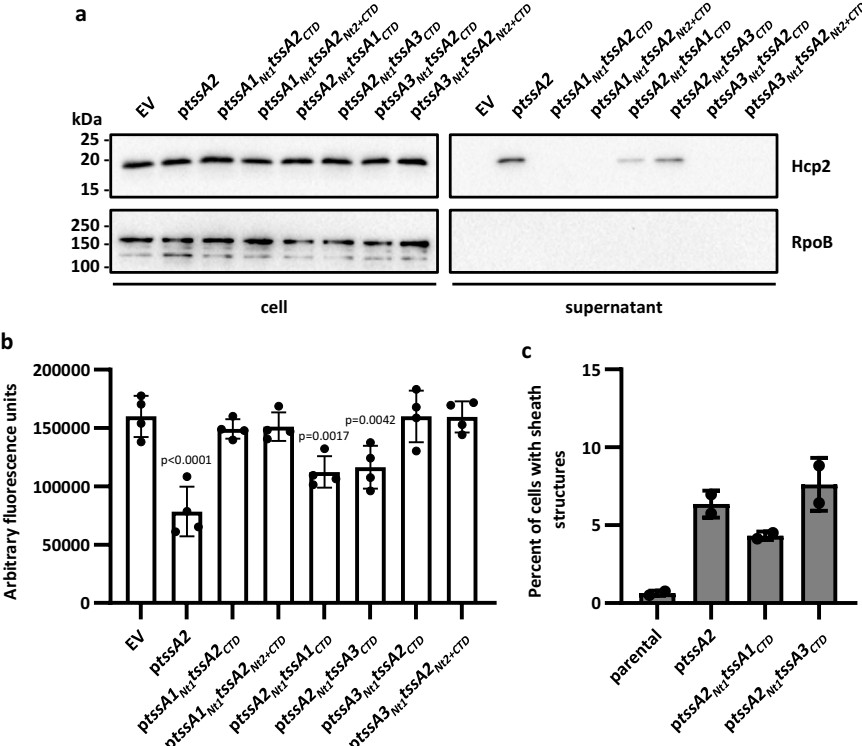

**Fig. 7 | Activity of H2-T6SS with chimeric TssA proteins.** To assess complementation of the *tssA2* deletion, the pBBR1MCS4 vector was introduced either as EV, encoding *tssA2* (p*tssA2*) or chimeric *tssA*s. **a** Western blot of a H2-T6SS secretion assay in the PAO1Δ*rsmA*Δ*tssA2* background to assess the levels of secretion marker Hcp2 in the supernatant sample. RpoB was used as a bacterial lysis control. Representative of 3 repeats. **b** H2-T6SS competition assay to assess the T6SS-dependent killing of a GFP-encoding *E. coli* prey by *P. aeruginosa* attacker strains in the PAO1Δ*rsmA*Δ*tssA2* background. (*n* = 4 independent experiments). **c** Quantification of the percentage of cells with TssB2-mScarlet sheath structures seen by fluorescence microscopy in the PAO1Δ*rsmA*Δ*tssA2 tssB2-mScarlet-I*

background. Sheath structures were formed in 0.64% (*n* = 9920 cells examined) of cells in the absence of *tssA2*, upon introduction of *tssA2* on a plasmid this rose to 6.30% (*n* = 5412 cells examined). Upon introduction of chimeric *tssA2*<sub>Nt1</sub>*tssA1*<sub>CTD</sub> or *tssA2*<sub>Nt1</sub>*tssA3*<sub>CTD</sub>, 4.42% (*n* = 4687 cells examined) and 7.67% (*n* = 5387 cells examined) of cells formed sheath structures respectively. Data was taken from three fields of view for two experiments. Statistical testing was conducted by one-way ANOVA with Dunnett's multiple comparisons test, each strain was compared to the parental EV strain. Values are presented as means, error bars represent standard deviation. Source data are provided as a source data file.

## Model of TssA interaction with the TssBC sheath

Considering our data, we subsequently investigated whether docking models between TssA and the sheath could corroborate a role for the Nt1 domain hairpin/loop regions in the specificity of interaction (Fig. 9). There are currently no T6SSs where structures for both the sheath and the TssA Nt1 domain are known. For *P. aeruginosa*, the structure of the H1-T6SS sheath (TssB1C1) has been resolved in its contracted conformation, where TssA1 has been observed to bind to one end of this structure[13,39]. Here, we use the zDock server[53] to generate a model for the TssA1 Nt1 domain docked to the distal end of the contracted TssB1C1 sheath structure (Fig. 9a). An equivalent docking using a model for the extended TssB1C1 sheath structure revealed a similar predicted positioning of the Nt1 domain at the distal end of this structure (Supplementary Fig. 14). One area that was identified to be variable between TssA1 and TssA3 Nt1 domain models was the loop region, which is found between α1 and α2 of the Nt1 domain, within the ImpA_N region[38,54]. As expected, due to similar loop sizes, docking analysis of TssA3 with the H1-T6SS sheath did not reveal any steric clash, indicating that the lack of interaction relates to amino acid sequence (Fig. 9b).

Downstream of the ImpA_N region, a hairpin region was also identified to be variable between TssA1 and TssA3. In the docking of the TssA1 Nt1 domain to the H1-T6SS sheath structure, the hairpin region is identified to fit within a groove. However, equivalent docking of the TssA3 Nt1 domain introduced a steric clash with the H1-T6SS sheath (Fig. 9c). Steric hindrance may explain why neither

the TssA3 Nt1 domain nor TssA1 with the TssA3 hairpin region interact with H1-T6SS sheath components and support our hypothesis that there is a structural constraint that contributes to TssA functional specificity.

## Discussion

Here we have shown that despite structural differences between TssA C-terminal regions, these could be functionally exchanged in a heterologous context. Instead, it is the more conserved Nt1 domain, which is essential in functional chimeras, and despite similarities, engages highly specific interactions with the cognate TssBC sheath. This clearly implicates Nt1 domains in recruitment of TssA to the T6SS and in previously described TssA roles associated with coordinating the polymerisation of the T6SS tail complex.

It was previously identified that the TssA<sup>EC</sup> interaction with the sheath involved a region comprising the Nt1 and Nt2 domains[19]. Based on this, the Nt2 domains of TssA<sup>EC</sup> and *Vibrio cholerae* TssA<sup>VC</sup> have been modelled to dock to the distal end of the sheath, interdigitating with subunits which was suggested to stabilise the extended conformation of the sheath during its polymerisation[19,37]. As demonstrated here and elsewhere, TssA<sub>S</sub> proteins also interact with sheath components despite lacking an Nt2 domain[38,39]. We identified that the Nt1 domains of both long and short TssA proteins specifically interact with their cognate sheath proteins, indicating a conserved mechanism for sheath interaction between these diverse structures. Structural similarity has been identified between the N-terminal portion of the *B. cenocepacia*

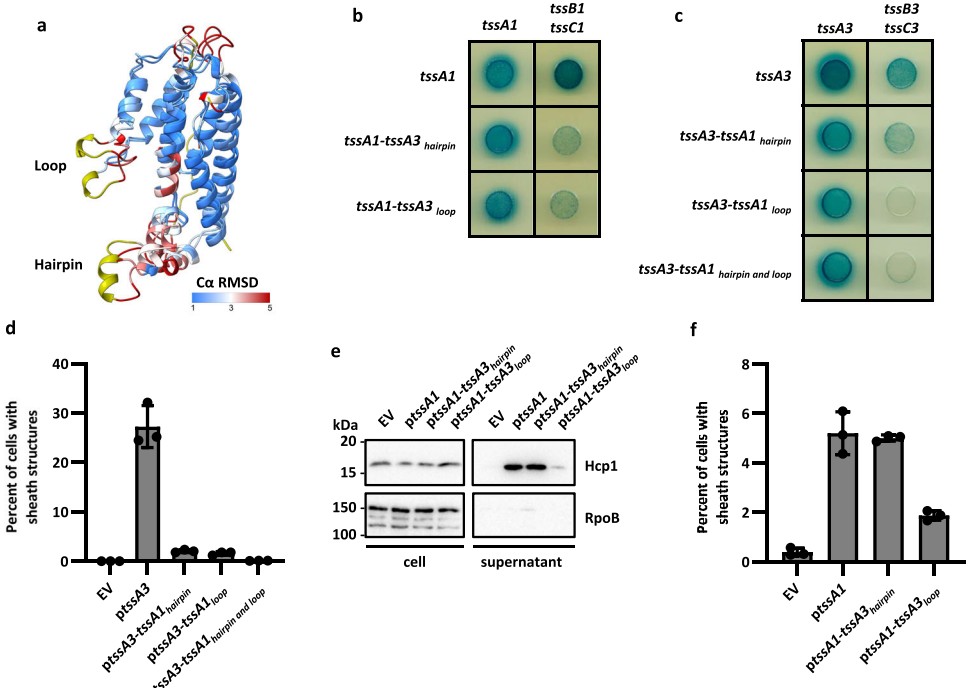

**Fig. 8 | Exchange of TssA1 and TssA3 hairpin and loop regions disrupts TssA function. a** Alignment of TssA1 and TssA3 Nt1 domain models. TssA Nt1 and TssA3 Nt1 homology molecular models were structurally aligned to minimise the root mean square deviation (RMSD) between their corresponding Cα atoms. Super-imposed structures were coloured according to the RMSD of Cα atoms. Blue indicates regions where the Cα atoms are within 1 Å of each other in the two models. Red highlights regions where the Cα atoms are 5 Å or more apart in the corresponding proteins. Yellow highlights regions where the two sequences differ, showing no correspondence between the residues due to the presence of an inserted or deleted region. **b** BTH interactions of TssA1 with TssA3 hairpin or loop regions with TssA1 and H1-T6SS sheath components. TssA1-TssA3$_{hairpin\ and\ loop}$ was not stable and therefore discarded. **c** BTH interactions of TssA3 with TssA1 hairpin and/or loop regions with TssA3 and H3-T6SS sheath components. **d** Quantification of the percentage of cells with TssB3-mScarlet sheath structures by fluorescence microscopy in the PAO1Δ*rsmA*Δ*tssA3 tssB3-mScarlet-I* background. To assess complementation of a *tssA* deletion, the pBBR1MCS4 vector was introduced either as EV, encoding *tssA1* (p*tssA1*), *tssA3* (p*tssA3*) or *tssAs* with exchanged hairpin and/or loop sequences. Sheath structures were formed in 0.04% (*n* = 16822 cells examined)

of cells in the absence of *tssA3*, this rose to 27.23% (*n* = 16253 cells examined) with expression of *tssA3* on a plasmid. Upon introduction of *tssA3-tssA1$_{hairpin}$*, *tssA3-tssA1$_{loop}$* and *tssA3-tssA1$_{hairpin\ and\ loop}$* sheath structures were detectable in 2.31% (*n* = 10760 cells examined), 1.60% (*n* = 13968 cells examined) and 0.08% (*n* = 11530 cells examined) of cells respectively. Data was taken from three fields of view for two experiments. **e** Western blot of a H1-T6SS secretion assay in the PAO1Δ*rsmA*Δ*rsmN*Δ*tssA1* background to assess the levels of secretion marker Hcp1 in the supernatant sample. RpoB was used as a bacterial lysis control. Representative of 3 repeats. **f** Quantification of the percentage of cells with TssB1-mScarlet sheath structures by fluorescence microscopy in the PAO1Δ*retS*Δ*tssA1 tssB1-mScarlet-I* background. Sheath structures were formed in 0.37% (*n* = 10017 cells examined) of cells with *tssA1* deletion, upon introduction of *tssA1* on a plasmid this rose to 4.95% (*n* = 8117). Upon introduction of *tssA1-tssA3$_{hairpin}$* 5.02% (*n* = 10683 cells examined) of cells formed sheath structures, this was 1.87% (*n* = 9471 cells examined) for introduction of *tssA1-tssA3$_{loop}$*. Data was taken from three fields of view for two experiments. Values are presented as means, error bars represent standard deviation. Source data are provided as a source data file.

TssA$^{BC}$ Nt1 domain and the *Aeromonas hydrophila* TssA$^{AH}$ Nt2 domain, which may explain why the Nt2 domain could be docked to the sheath structure when it appears that Nt1 domains are involved in this interaction[38]. This N-terminal portion of the Nt1 domain contains the highly conserved ImpA_N region, which is therefore implicated in sheath interaction. An ImpA_N region is also present in the TssA$_L$-associated accessory protein TagA, which is proposed to localise with the distal end of the polymerised sheath, stabilising its extended conformation[18]. Interaction between TagA and the sheath, as identified in *V. cholerae*[41], may likewise involve the ImpA_N region.

Given the sequence conservation in the Nt1 domain between TssA proteins, particularly in the ImpA_N region, the functional conservation of sheath binding is not surprising. The C-terminal regions of TssA proteins are far more variable, particularly between long and short TssA proteins, but variations are also apparent within these groups, with an extension to the TssA$_L$ C-terminal domain identified to give rise to a sub-group with distinct structural features[38]. Despite the structural diversity of C-terminal domains, functionality is conserved, with each of these domains responsible for oligomerising TssA ring structures. This diversity does, however, introduce variation in the dimensions of these rings. The oligomerisation of the C-terminal domain of the short

TssA$^{BC}$ forms a large ring with a lumen of 100 Å and an external diameter of 200 Å[38]. In contrast, the oligomerisation of the C-terminal domain of TssA$_L$ proteins forms a smaller ring with a 25–50 Å lumen and an external diameter in the region of 100–130 Å with dimeric Nt2 domains flexibly associated with the periphery[19,37,38]. It is noteworthy that the diameter of the TssA$_L$ ring is increased by the presence of these Nt2 domains, forming a thicker ring structure with diameter similar to the 200 Å observed for TssA$_S$[37,38]. It was previously suggested that these distinct ring structures may function as a platform that facilitates the proper positioning of the more conserved Nt1 domains at an equivalent radial position[38]. This theory is supported by the activity of chimeric TssA proteins upon the exchange of both very similar C-terminal regions, such as those of the short TssA1 and TssA3, and distinct C-terminal regions, such as that of the long TssA2 for those of the short TssA1 or TssA3. While specific C-terminal regions were not required for T6SS activity, not all chimeric TssA proteins were functional, indicating that features regulated by the C-terminal domain, such as symmetry and ring dimensions, may contribute to TssA activity in ways that remain to be characterised.

The theory of a conserved mechanism for interactions between TssA and the sheath is complicated slightly by the variability in TssA

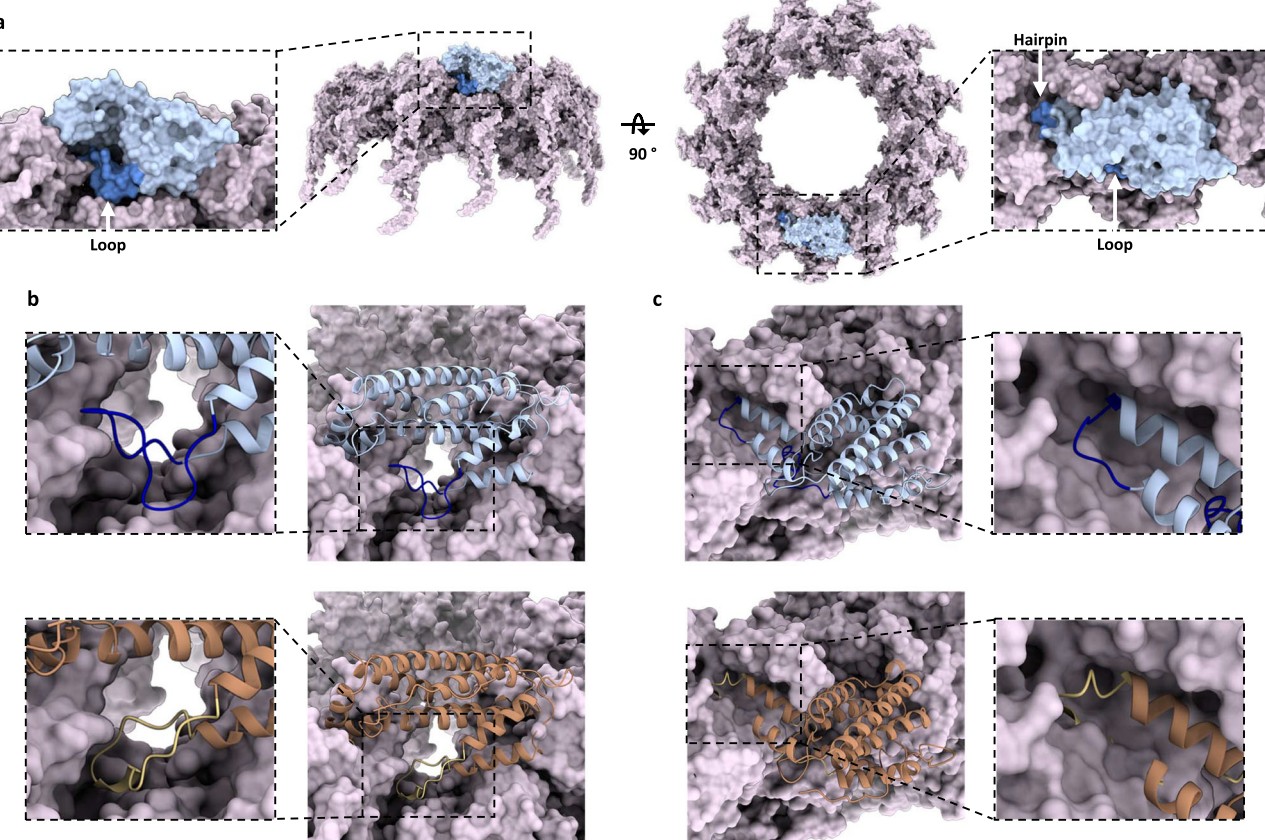

**Fig. 9 | Docking of the Nt1 domain at the distal end of the sheath. a** Model for the docking of the TssA1 Nt1 domain (light blue) at the distal end of the contracted sheath (light purple) (PDB:5N8N). Side and top views are given. The position of TssA1 Nt1 domain hairpin and loop regions (blue) are highlighted. **b** Position of the TssA1 (light blue) and TssA3 (orange) Nt1 domain loop regions when docked to the H1-T6SS sheath. The TssA1 and TssA3 loop regions are highlighted in blue and gold respectively. **c** Position of the TssA1 and TssA3 Nt1 domain hairpin regions when docked to the H1-T6SS sheath. The TssA1 hairpin region (blue) fits within a groove. The extended TssA3 hairpin region (gold) sterically clashes with the sheath structure.

symmetries. The contractile sheath is assembled with six-fold symmetry, which matches the six-fold symmetries reported for certain long TssA proteins, TssA[EC] and TssA[VC], but would clash with the five-fold and 16-fold symmetries reported for the long TssA[AH] and short TssA[BC], respectively. To date, no structure of a Nt1 domain connected to another domain has been solved at a high resolution, which has been attributed to a high degree of flexibility in the interdomain linker[37–39]. Flexibly associated at the periphery of TssA ring-shaped structures through these linkers, Nt1 domains are therefore likely not to be constrained by the associated ring symmetries, similar to the flexibility identified for Nt2 domains in relation to their C-terminal domain rings[37,38]. This flexibility would allow Nt1 domains to interact with their respective sheaths of six-fold symmetry independent of the variable symmetries of their C-terminal domain rings. It is tempting to propose that all interactions of TssA proteins with T6SS components are mediated through the Nt1 domains, both because of their conservation and because their flexibility relieves them of the symmetry constraints associated with the ring structure. However, there are indications from previous studies of interactions between C-terminal domains and T6SS components, such as VgrG and Hcp proteins, indicating that other contributions of C-terminal domains should be considered[19,38,41].

The general model for TssA localisation during T6SS assembly involves static TssA associated with the site of the membrane complex and baseplate complex, becoming dynamic upon polymerisation of the tail complex, where it is bound at the growing distal end of the sheath as it extends across the cell to the opposite membrane[19,37,38,42].

From this position, a single TssA complex remains associated and is proposed to coordinate the polymerisation of the growing tail complex, although the mechanism by which new sheath and Hcp subunits are incorporated with TssA bound is currently unclear[19,55,56]. A cap model was previously proposed whereby a TssA complex at the distal end of the tail complex maintains contact with the sheath as new subunits are incorporated. This resembles the capping of the flagellar tip by FliD/HAP2, where an oligomerised platform sits atop a polymerising helical structure with peripheral domains that interact with this structure and coordinate polymerisation[57]. A further interesting parallel is that, like TssA, variable FliD symmetries have been identified (four-, five- and six-fold) which also clash with the 11-fold symmetry of the flagellar filament[57–63].

We now propose an updated cap model, building on our discovery that sheath interaction involves TssA Nt1 domains. A molecular docking approach proposed that Nt1 domains fit into grooves at the surface of the distal end of the sheath (Fig. 9a). With all TssA complexes described to date comprised of at least 10 monomers, a maximum of six Nt1 domains would be expected to bind these grooves, with remaining Nt1 domains being unbound. We propose that free Nt1 domains could bind new TssBC protomers and recruit them to the polymerising sheath (Fig. 10). In this model, as in the proposed model for flagellar polymerisation, the addition of a new sheath subunit would introduce a conformational change that displaces a neighbouring Nt1 domain from its groove. In such a mechanism, the remaining Nt1 domains would maintain contact with the sheath with sequential Nt1 domain displacement and polymerisation of sheath subunits.

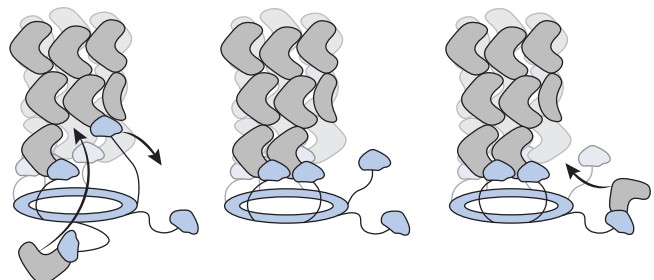

**Fig. 10 | Schematic for an updated cap model for sheath polymerisation.** A TssA complex (light blue), formed of a ring structure with flexibly associated Nt1 domains, is bound at the distal end of the sheath (grey). For simplicity, a subset of Nt1 domains is represented, but in reality, many more would be flexibly associated with the ring. Interaction between TssA and the sheath is mediated by Nt1 domains which insert in binding grooves at the tip of the sheath. In this model, an unbound Nt1 domain can recruit a free sheath subunit, whose incorporation into the distal end of the sheath generates a conformational change that displaces a neighbouring TssA Nt1 domain subunit from its binding groove. Through this process, new sheath subunits are incorporated while TssA remains bound.

Overall, unravelling the details of the assembly and contraction of the T6SS sheath requires a clear understanding of how the different components of the system can be pieced together and sustain a dynamic rearrangement. Our work on the Nt1 domain and how it specifically fits into the sheath structure is instrumental to this. We also provide evidence for how function and positioning of very diverse TssA structures can be conserved through the orientation of these Nt1 domains by structurally distinct, but functionally similar, C-terminal regions.

## Methods

### Bacterial strains, plasmids and growth conditions
Unless otherwise stated, LB broth (Miller) and agar (Miller) were used for routine growth of all organisms at 37 °C with 150 rpm orbital shaking of liquid cultures. Bacterial cultures for secretion assays and microscopy experiments were grown using tryptone soya broth (TSB) (Sigma). Growth media were supplemented with the following, as required: 100 µg/mL ampicillin, carbenicillin 250 µg/ml, gentamicin 50 µg/ml, 0.5-1 mM Isopropyl β-D-1-thiogalactopyranoside (IPTG), 50 µg/mL kanamycin, 100 µg/ml streptomycin for *E. coli* or 1000–2000 µg/ml streptomycin for *P. aeruginosa*, 100 µg/ml 5-Bromo-4-chloro-3-indolyl beta-D-galactopyranoside (X-Gal).

A list of primers used in this study is present in Supplementary Table 1, bacterial strains used in this study are present in Supplementary Table 1 and plasmids used in this study are listed in Supplementary Table 3. *P. aeruginosa* PAO1 chromosomal mutants were constructed by allelic exchange as described previously[64]. Briefly, a mutator fragment containing the desired sequence alteration surrounded by >500 bp regions of sequence homologous to the chromosome upstream and downstream was introduced into the pKNG101 suicide vector[65]. By three-partner conjugation, using the pKNG101-carrying *E. coli* CC118λpir donor strain[66] and *E. coli* 1047 pRK2013 helper strain[67], the suicide vector was transferred to the receiver *P. aeruginosa*. Chromosomal integration was selected for with 1000 µg/ml streptomycin, followed by counterselection with 20% (w/v) sucrose.

### Bacterial two-hybrid
Briefly bacterial two-hybrid (BTH) constructs were generated by cloning genes of interest in frame into the bacterial two-hybrid vectors (pKT25 or pUT18C)[46], giving rise to either N- or C-terminal fusions to the T18 or T25 subunits of the catalytic domain of *Bordetella pertussis* adenylate cyclase, CyaA. One T18- and one T25-encoding plasmid were co-transformed into *E. coli* DHM1 competent cells lacking an endogenous adenylate cyclase. In triplicate, single colonies were inoculated

into LB broth supplemented with appropriate antibiotics and 0.5 mM IPTG and grown at 30 °C overnight. Culture was spotted onto LB agar supplemented with appropriate antibiotics, 1 mM IPTG and 100 µg/ml X-Gal and incubated at 30 °C for 24 or 40 h. Protein-protein interaction reconstitutes a functional adenylate cyclase, which can drive β-galactosidase production allowing X-Gal hydrolysis, giving rise to a blue colorimetric output.

### Protein expression and purification
TssA C-terminal domains were expressed from pETDuet-1 MCS1 in *E. coli* BL21(λDE3). Cells were grown in 1 L of LB broth supplemented with appropriate antibiotics at 37 °C until reaching an $OD_{600}$ 0.6, whereupon 0.5 mM IPTG was added. The culture was then incubated at 20 °C with agitation overnight. Cells were harvested and resuspended in 30 ml lysis buffer per 1 L of culture [20 mM N-2- hydroxyethylpiperazine-N-2-ethane sulfonic acid (HEPES) pH 7.5, 250 mM NaCl, 100 µl/100 ml Triton x100, 0.5 mM phenylmethylsulfonyl fluoride (PMSF), 1 mg/ml lysozyme]. Cells were sonicated and then centrifuged (195,000 × *g*, 4 °C, 45 min) to clarify the cellular lysate. The supernatant was passed through a 0.45 µm syringe filter (Sartorious) and then loaded onto a Strep-trap HP 1 ml column (Sigma) using an ÄKTA purifier (GE Healthcare). The column was washed, and the protein eluted (PBS containing 2.5 mM desthiobiotin). The eluted sample was dialysed (20 mM HEPES pH 7.5, 150 mM NaCl) and further purified on a Superose 6 10/300 GL column (GE Healthcare) using 20 mM HEPES pH 7.5, 150 mM NaCl as the running buffer.

### Electron microscopy sample preparation, data collection and image processing
To assess the different TssA variants protein purity and homogeneity by negative-stain (NS) electron microscopy, samples were diluted to 0.003 mg/ml with dialysis buffer (20 mM HEPES pH 7.5, 150 mM NaCl) and 10 µl was applied onto glow-discharged carbon-coated copper grids (300 mesh, Agar Scientific) and incubated for 2 min at room temperature. Grids were held with clamping forceps, washed with water twice for 10 seconds and the excess water was blotted off onto filter paper (Whatman). Grids were then stained with 2% (w/v) uranyl acetate for 20 s, the excess of stain was blotted off and grids were left to dry before imaging. Negative stain images were acquired on a FEI Tecnai G2 Spirit operating at 120 kV equipped with a LaB6 filament and a CCD 4k × 4k camera.

For the cryo-electron microscopy (cryo-EM) imaging, 4 µl of the $TssA1_{CTD}$ sample (adjusted to 0.1 mg/ml) was applied onto glow-discharged Quantifoil R 2/2 on 200 gold mesh. Sample was vitrified in liquid ethane using a Leica automatic plunge freezer EM GP2 with a 2.4–2.8 s blot time under 85% relative humidity at 4 °C. The frozen grids were checked for sample concentration and ice thickness using a Jeol 2100 plus microscope equipped with a Gatan One View detector. Images were collected at a magnification of 50.000× corresponding to a pixel size of 2.16 Å/pixel.

The CTF of the collected images was estimated using the patch CTF estimation function in CryoSPARC and after screening the micrographs for good Thon Rings, 16577 particles ($TssA1_{CTD}$) and 4904 ($TssA3_{CTD}$) were automatically picked using the blob picker function. Particles were extracted from the micrographs and classified iteratively until generating interpretable 2D classes.

### Secretion assay
*P. aeruginosa* strains carrying pBBR1MCS4[68] vectors were grown overnight at 37 °C in TSB supplemented with carbenicillin. Bacteria were then sub-cultured to an $OD_{600}$ 0.1 in 20 ml TSB. For the H1-T6SS, cultures were grown at 37 °C for at least 5 h, for the H2-T6SS cultures were grown at 25 °C for 18–24 h. Following this incubation, a whole cell lysate sample was pelleted and resuspended in Laemmli buffer to a concentration of 0.1 $OD_{600}$ equivalent per 10 µl. To isolate an

extracellular supernatant fraction, the culture was sequentially centrifuged (4×, 4 °C, 3400 × g for 20 min) to separate the supernatant from any remaining cells. Proteins were precipitated from the supernatant with 10% trichloroacetic acid (TCA) (Sigma), before being washed with 90% acetone. The recovered pellet was air-dried before addition of Laemmli buffer to a concentration of 1.0 $OD_{600}$ equivalent per 10 μl. Whole cell and supernatant samples were then resolved by SDS-PAGE before being visualised by western blot to identify the presence of RpoB and the relevant Hcp. Antibodies used in this study[27,43,69] are provided in Supplementary Table 4.

### Interbacterial competition assays
Attacker *P. aeruginosa* strains carrying pBBR1MCS4[68] vectors and prey *E. coli* carrying pRL662-GFP[7] were grown overnight at 37 °C in LB supplemented with appropriate antibiotics. Cellular pellets isolated and washed in PBS before adjusting to an $OD_{600}$ of 3.0. Attacker and prey were mixed at a 1:1 ratio, before being centrifuged and half the supernatant removed. In triplicate, 5 μl competition spots were aliquoted onto well dried LB agar plates. CFU of the input bacteria were checked to confirm the ratio of prey and attacker in each competition. For H1-T6SS competitions, competition mixes were incubated at 37 °C for 24 h, for H2-T6SS competitions, competition mixes were incubated at 25 °C for 24 h. Following incubation, competition mixes were recovered into 1 ml sterile PBS, vortexed and the GFP fluorescence intensity, representing prey *E. coli* pRL662-GFP abundance, was measured using a microplate reader (BMG Labtech). Technical triplicates were averaged to give one biological replicate. Data presented are the means and with error bars representing the standard deviation of at least three biological replicates. Multiplicity adjusted P values are reported with a familywise alpha threshold of 0.05.

### Fluorescence microscopy
For imaging the H3-T6SS, samples were prepared for microscopy imaging by growth in liquid culture as described previously[36]. Briefly, strains were grown at 37 °C in TSB supplemented with appropriate antibiotics before being sub-cultured to an $OD_{600}$ of 0.1 in 50 ml TSB without antibiotic. Bacteria were grown at 25 °C for 13 h with shaking at 200 rpm to an $OD_{600}$ between 3.0–5.0. Cells were then imaged using a glass bottomed 35 mm imaging dish (μ-Dish 35 mm, high Glass Bottom) (ibidi). 6 μl of bacterial culture was transferred to the glass base of the dish and a Vogel Bonner Medium (VBM) agar pad was placed on top of the culture to sandwich it between the agar and the glass.

For imaging the H1- and H2-T6SS, samples were prepared for microscopy imaging by growth on solid media. Strains were grown at 37 °C in TSB supplemented with appropriate antibiotics before harvesting cells by centrifuging at 5500 × g for 3 min. The pellet was washed with sterile PBS and the $OD_{600}$ adjusted to 0.05. 6 μl of diluted culture was then spotted onto an LB agar pad and well dried. This agar pad was then inverted and transferred to a glass bottomed 35 mm imaging dish where the dried spot was sandwiched between the agar and the glass, as above. For the H1-T6SS samples were incubated at 37 °C for 3–4 h, for the H2-T6SS samples were incubated at 25 °C for 5–6 h.

Samples were imaged using the Axio Observer Z1 (Zeiss) inverted widefield microscope with a 100×/1.4 Oil Ph3 M27 PlanApochromat objective (Zeiss), a SpectraX LED light engine (Lumencore) and an ORCA-Flash 4.0 digital CMOS camera (Hamamatsu). An environmental control system was used to regulate the temperature of the sample within the microscope chamber. Images were captured in parallel using phase contrast to assess total bacterial abundance and fluorescence to capture signal from mScarlet-I fluorophores. Microscopy was performed at Facility for Imaging by Light Microscopy (FILM) Imperial College London.

Microscopy images were analysed using ZEISS ZEN 3.4 (blue edition) and FIJI ImageJ version 2.0.0/1.53c[70]. FIJI was used to determine cell count and to count fluorescent foci.

### Generation of C-terminal HA-tagged expression vectors
C-terminal HA tag was inserted into pBBR1MCS4 expression vectors carrying the different chimeras using the Q5 Site Directed Mutagenesis kit (NEB) following manufacturers protocol. Briefly, using custom mutagenic primers, PCR of the template plasmid was carried out to insert the HA tag at the C-terminal end. After PCR, the amplified material was treated with a Kinase-Ligase-DpnI mix to enrich for mutated DNA and transformed into NEB 5-alpha competent *E. coli* cells. The HA-tagged expression vectors were then verified by Sanger sequencing to confirm the presence of the tag. Samples were prepared for western blot analysis as described for secretion assay whole cell extract samples.

### Protein homology molecular modelling and analysis
Protein structural models of TssA Nt1 domains were generated using Robetta protein structure prediction service[71] and SWISS MODEL protein structure homology-modelling server[72]. GetArea was used to calculate the solvent accessible surface area of proteins[51]. Molecular docking of proteins was performed using the ZDock server[53]. Protein structural alignment was carried out using the Dali protein structure comparison server[52].

### Reporting summary
Further information on research design is available in the Nature Portfolio Reporting Summary linked to this article.

## Data availability
All data are available in the main text and supplementary materials, with uncropped scans of all gels and blots, as well as raw data for all graphs, provided in source data files. The structural data of the *P. aeruginosa* contracted sheath used in this study are available in the Protein Data Bank (PDB) under accession code 5N8N. Source data are provided with this paper.

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

## Acknowledgements
This work was supported by the MRC grant MR/S02316X/1 awarded to A.F. and an MRC studentship awarded to S.F.

## Author contributions
A.F. and S.F. conceived the investigation. S.F., P.P., A.I., S.S., C.A.Z.T and T.R.D.C. performed the experiments. A.F., S.F., P.P. and T.R.D.C. designed the experiments, interpreted the data, and wrote the manuscript.

## Competing interests
The authors declare no competing interests.
