## [Peer Review File · Nature Communications]

Functionality of chimeric TssA proteins in the type VI secretion system reveals sheath docking specificity within their N-terminal domainsReviewer #1 (Remarks to the Author):

In this manuscript the authors analyze TssA proteins from *P. aeruginosa* and show that the N-terminal domains are required for specific interaction with the cognate sheaths and the C-terminal domains are required for oligomerization of the respective TssA. The authors convincingly show the role of Nt1 domains in establishing specificity of binding between TssA proteins and their corresponding sheath binding partners. Moreover, they show that the respective C-terminal domains can be swapped in some cases and preserve T6SS function. Text is well written and data seem solid.

Major issues:

Could authors explain why they show in Figure 9 a model of TssA interacting with a contracted sheath? Fitting of TssA to an extended sheath is only shown in Supplementary Figure 9 but it is unclear how was that extended structure generated. Would fitting different TssAs change if the extended structure is used and would this influence the explanation why TssA3 cannot interact with H1-T6SS sheath?

Although the authors show convincingly that certain chimeric versions of TssA are able to restore T6SS function and killing, some visualization of the assembly dynamics should be provided. In particular this applies to experiments with the TssA2 Nt1 chimera with short TssA which is shown to have intermediate function, as well as loop and hairpin chimeras (Figure 8). Considering the model proposed by the authors of T6SS assembly, it would be important to see if the non-functional chimeras fail to form any structures at all or if there are aborted assemblies. In general, authors should consider showing examples of microscopy images that were analyzed to obtain the quantitative data shown in figures.

In Figure 2b, it appears that killing in the Δ tssB1 EV strain is lower than in the Δ tssA1 EV strain, suggesting some assemblies still occur in the Δ tssA1 EV strain. Moreover, data in Figures 6c and 7c suggest that some assemblies occur in the absence of the cognate TssA. Please, show example images or videos showing those assemblies and provide a discussion of how assembly might happen in the absence of the cognate TssAs.

Minor issues:

Fig. 1 and Fig. 5 could be moved to supplements.

As the authors propose a general mechanism for T6SS TssA protein function, acting as a loading scaffold during assembly, it would be interesting to test whether C-terminal domains from other species (especially TssA short variants) would also complement function in *P. aeruginosa*.

While obvious to readers familiar with the used technique, it would be good to include some explanation for the color changes observed in the BTH experiments.

Loop and hairpin comparisons for TssA1 and TssA3 are missing RMSD values.

Reviewer #2 (Remarks to the Author):

This study focuses on a description of chimeric TssA proteins produced by *Pseudomonas aeruginosa*. TssA are gp6-like key oligomeric components of the type 6 secretion system hexagonal inner baseplate, which are involved in sheath polymerization. The main findings are (i) TssA are modular proteins with two domains, (ii) the N terminal domain binds to sheath proteins via two regions, (iii) the C terminal domain control ring formation/oligomerization and can be swapped from TssA subgroups.

The work is incremental or at least confirmatory of a set of previous studies. Some of the data presented here is similar to data published by others.

- Figure 3: The interaction of TssA1 with TssBC proteins has been shown in previous studies including one of the authors themselves (Planamente et al 2016). The requirement for domain Nt1 is also documented (Zoued et al. 2016).

- Figure 4: Structural information on oligomeric states of TssA subgroups, and the contribution of the C terminal domains in association has been already provided with higher resolution (Dix et al. 2018)

- Figure 9: TssA-sheath molecular docking simulations have been published (Schneider et al 2019) and the author's model of the role of TssA for sheath polymerization has been proposed (Schneider et al 2019, Zoued et al 2016). What is surprising is that the authors model TssA bound to the distal end of the sheath while they showed that is a component of the baseplate, likely to be located at the basal end of the sheath.

Noteworthy the lack of scholarship at several instances in the manuscript is blatant, e.g. "A remarkable feature of TssA proteins is the ability to form multimers, which was PROPOSED to occur through the C terminal domain^{41,42}". But this was DEMONSTRATED! In Ref 41 the structures of various TssA were reported demonstrating the role of the C terminal domains in oligomerization. And other structures demonstrating this were published in references 19, 21, and 40.

The novelty of this work resides and is therefore limited to (i) the observation that TssA binding to the sheath requires the two sheath subunits and two discrete regions of TssA and (ii) to swapping experiments demonstrating the C terminal is functionally permissive.

Some essential controls are missing, eg the authors use the BTH assay to conclude about the stability of their chimeric constructs, however in the BTH assay the chimeras are fused to stable domain and are produced in *E. coli*, which does not reflect the in vivo case, in *P. aeruginosa*. Statistical analyses are missing (Fig 2).

There is a number of discrepancies with a previous study from the same authors, eg Planamente et al 2016 showed that "TssA1 is not strictly essential for TssB1C1-sheath assembly" while it seems essential in this study.

In addition the authors previously showed that *Pseudomonas* TssA1 has topology/structural similarities with bacteriophage gp6 and is likely to be a structural subunit of the type 6 secretion baseplate interacting with the first layer of sheath (Planamente et al 2016: "The existence in the T6SS of a gp6-like domain that we describe here within TssA1 is a breakthrough since gp6 is central to the phage baseplate organization."), and this is not discussed in this work. They showed that TssK is an gp8 homologue (Planamente et al 2016) and interactions between TssA domains and the gp8 homologue is not tested here. The gp6 inner baseplate ring is asymmetric (Taylor et al 2016) and is comprised of 12 copies. The comparison between gp6 and TssA oligomeric states and symmetry could be discussed here. This is not clear to me as the authors mention in their 2016 article that "It is thus clear that the TssA protein that we describe in our study, TssA1, and EcTssA (or TssA2) are two distinct proteins and most of the hypothetical mechanism that has been associated with EcTssA remains to be experimentally validated in other T6SSs." It seems to me that the authors now suggest that their TssA1 protein has a function similar to EcTssA and therefore they validate the EcTssA hypothetical mechanism. This should be discussed in more details and if TssA is not a gp6-like baseplate protein this should be clearly mentioned and corrected in this work.

Reviewer #3 (Remarks to the Author):

In this paper, Fecht et al compared the TssA proteins among three T6SSs in *Pseudomonas*

aeruginosa. Through genetic mutation and complementation experiments, the authors found that the TssA proteins were specific to their corresponding T6SSs by interacting with cognate TssB/C sheath subunits. By constructing different TssA chimeras, the authors demonstrated that the Nt1 domain of TssA played a pivotal role in this process. Further comparison of the Nt1 domains of TssA1 and TssA3 revealed the presence of specific loop and hairpin secondary structures. Overall, this study provides valuable information for understanding the specific interactions between the TssA protein and the TssB/C sheath. However, some of the conclusions need more experimental support.

Major issues:

1. Figure 2b: why did the tssA1 mutant exhibit stronger T6SS-mediated killing abilities compared with the tssB1 mutant? Besides, for the complementation experiments, the author should validate the expression of those TssA proteins and the TssA chimeras.
2. For the cryo-EM study of the C-terminal domain of both TssA1 and TssA3, more linearized shapes were observed in TssA3CTD compared to TssA1CTD; the authors mentioned this may be due to flexibility. Did the authors try other truncated versions of TssA3CTD to improve the stability? In the manuscript, the authors claimed that the TssA3CTD presents a 12-fold symmetry arrangement. However, in Fig. 4d and f, the resolution is poor and difficult to conclude.
3. For the long TssA, what's the role of the Nt2 domain? In the introduction, the authors mentioned the Nt2 domain tends to form dimers surrounding the C-terminal domain ring of TssAL. Does this Nt2 domain participate in the formation of the ring structure? When the authors investigated the structure of TssA2CTD by cryo-EM, did the authors try to include the Nt2 domain in the C-terminal domain for structural study?
4. The experimental design must be more convincing to prove that the hairpin and loop regions of Nt1 are responsible for the specificity in sheath interaction. For example, in Fig. 8b, the authors should include the detection of the interaction between TssA1-TssA3hairpin and loop and TssB1/TssC1. To make the conclusion more solid, the authors should perform the interaction assay with TssB3/TssC3. Similarly, the authors should conduct the binding assay of the TssA3-TssA1 variants with TssB1/TssC1 in Fig. 8c.
5. Figure 7: it is interesting that TssA chimeras lacking the Nt2 domain (TssA2Nt1-TssA1CTD and TssA2Nt1-TssA3CTD) were able to partially restore the H2-T6SS activities of tssA2 mutants. This raises the question of whether the Nt2 domain is essential for H2-T6SS activities. It would be interesting to test whether a TssA2 mutant lacking the Nt2 domain can also restore H2-T6SS activities in the tssA2 mutant or whether tssA2 mutants expressing TssA2Nt1-Nt2-TssA1CTD / TssA2Nt1-Nt2-TssA3CTD exhibit higher H2-T6SS activities.
6. Figure 8: the authors identified specific loop and hairpin secondary structures within the Nt1 domain of TssA1 and TssA3 that are essential for proper and specific interaction between TssA and the T6SS sheath. However, it remains unclear whether these secondary structures are unique to TssA1 and TssA3 or if they also exist in other TssA homologues and contribute to the interaction with the sheath. Further investigation into the conservation of these structural motifs across different TssA proteins would provide important insights into the mechanism of T6SS assembly.
7. Figure 9: the author proposed a docking model of TssA proteins with the sheath using the ZDock server. However, it is unclear why the TssA proteins can only interact with their cognate sheath but not with TssB or TssC independently. Additionally, the key residues or regions of TssB/C that contribute to the interaction with TssA were not identified. To address these gaps, the authors are recommended to test the interacting regions based on the current docking model in order to identify the specific regions of TssB/C that are responsible for the interaction with TssA.
8. The authors should revise Fig. 10 to make it more understandable.

Minor comments:

1. It is recommended that the authors consecutively number all lines of the manuscript for ease of reference.
2. *P. aeruginosa* contains three sets of T6SS, including H1-T6SS, H2-T6SS and H3-T6SS. To demonstrate each TssA is specific for its cognate T6SS, Hcp secretion and bacterial competition assays were performed for both TssA1 and TssA2, but the assembly of T6SS sheath structures was applied for TssA3. The authors should explain why these different methods were used.
3. There is a spelling mistake in the paper title, and it should be "specificity".
4. The format of references is not consistent, particularly for the title of the references. Also, the bacterial names and specific words like "in vivo" should be italic.

POINT BY POINT ANSWER TO REVIEWER'S COMMENTS

We are very grateful to the editors and reviewers for providing constructive comments and for giving us the opportunity to improve our manuscript. We have now addressed all the comments and revised the manuscript accordingly. We have paid particular attention to all main points requiring experimental approaches, which were to; i) engineering a new set of TssA chimera, including using TssA from a different species, namely *Pseudomonas putida*; ii) assessing functionality of the chimera using secretion and competition assays; iii) confirming that each chimera is produced at the same level (western blot analysis) and iv) broadening the TssA interaction with the gp8 homologue TssK. All changes made are described here below, under each reviewer comments which are highlighted in blue. Note that additional experiments, which are now included in the revised manuscript, have been conducted by Sujatha Subramoni and Casandra Ai Zhu Tan who have been included as a co-author. Also note that some of the main changes in the main text are highlighted in yellow.

REVIEWER'S COMMENTS

Reviewer #1 (Remarks to the Author):

In this manuscript the authors analyze TssA proteins from P. aeruginosa and show that the N-terminal domains are required for specific interaction with the cognate sheaths and the C-terminal domains are required for oligomerization of the respective TssA. The authors convincingly show the role of Nt1 domains in establishing specificity of binding between TssA proteins and their corresponding sheath binding partners. Moreover, they show that the respective C-terminal domains can be swapped in some cases and preserve T6SS function. Text is well written and data seem solid.

Major issues:

Could authors explain why they show in Figure 9 a model of TssA interacting with a contracted sheath? Fitting of TssA to an extended sheath is only shown in Supplementary Figure 9 but it is unclear how was that extended structure generated. Would fitting different TssAs change if the extended structure is used and would this influence the explanation why TssA3 cannot interact with H1-T6SS sheath?

We used the contracted sheath because this is a resolved structure, and there is no available structure for the extended form, and we felt that it is more reliable to dock the Nt1 model to a resolved structure than the model for the extended sheath. Additionally, we have previously demonstrated (Planamente *et al.*, 2016) that TssA interacts with the end of the sheath when produced in the contracted conformation.

Although the authors show convincingly that certain chimeric versions of TssA are able to restore T6SS function and killing, some visualization of the assembly dynamics should be provided. In particular this applies to experiments with the TssA2 Nt1 chimera with short TssA which is shown to have intermediate function, as well as loop and hairpin chimeras (Figure 8). Considering the model proposed by the authors of T6SS assembly, it would be important to see if the non-functional chimeras fail to form any structures at all or if there are aborted assemblies. In general, authors should consider showing examples of microscopy images that were analyzed to obtain the quantitative data shown in figures.

The reviewer is raising a fair and valid point. Yet, the dynamics of H2 assembly were incredibly slow, and sheaths were assembled for a very long time before contracting so that we could

not record any meaningful data in terms of sheath dynamics. In general, we did not identify any noticeable sheath defects across these strains with elongated sheaths detectable in all strains, although this cannot be ruled. It is also to be noted that we do not show any TssA2 data in figure 8.

We agree with the reviewer about showing examples of microscopy images that have been used for quantification. These have been assembled and now presented for all microscopy data in supplementary figures (Fig. S8 and Fig. S13), showing panels of images displaying sheath structures.

In Figure 2b, it appears that killing in the $\Delta tssB1$ EV strain is lower than in the $\Delta tssA1$ EV strain, suggesting some assemblies still occur in the $\Delta tssA1$ EV strain. Moreover, data in Figures 6c and 7c suggest that some assemblies occur in the absence of the cognate TssA. Please, show example images or videos showing those assemblies and provide a discussion of how assembly might happen in the absence of the cognate TssAs.

Assembly of T6SS sheath structures has been reported in the literature in the absence of TssA (e.g. in Schneider *et al.*, *EMBO J.* 38(18)e100825).

In terms of the difference between the *tssA* and *tssB*, we believe that the difference between mutants in this assay just demonstrates the difference between a component that is absolutely required (no T6SS activity in the absence of sheath) compared to one that might be required for an effective/optimal level of activity. One may assume that there can be some structures that form by chance without TssA, but it is infrequent, and the T6SS assemblies are not as effective. TssA would increase effectiveness of nucleating the sheath, while in absence of TssA it can still happen but is rarer and less effective. It is also important to recall that sheath structures could be obtained by simply overproducing TssBC in *Escherichia coli*, and this in the complete absence of any other T6SS components.

Where assembly of T6SS sheath structures has been reported in the absence of TssA (e.g. in Schneider *et al.*, *EMBO J.* 38(18)e100825), it was identified that they are mostly foci rather than dynamic structures. Similar is true in our case, where sheath structures reported in the absence of TssA proteins are primarily foci. We interpret this as possible examples of recruitment of sheath components to the baseplate, but failure to polymerise effectively without TssA. However, it obviously cannot be ruled out that these are extended sheaths viewed from an end rather than from the side. Therefore, we have reported all fluorescent structures as sheath structures for completeness, although we predict that many foci will not be competent to form an extended sheath.

Minor issues:

Fig. 1 and Fig. 5 could be moved to supplements.

We believe, Fig. 1 and Fig. 5 provide a good visual to clearly understand at one glance the organisation of TssA domains, oligomerisation and the rationale behind engineering all of the TssA chimera. As such we believe it is added value for it to be kept in the main text.

*As the authors propose a general mechanism for T6SS TssA protein function, acting as a loading scaffold during assembly, it would be interesting to test whether C-terminal domains from other species (especially TssA short variants) would also complement function in *P. aeruginosa*.*

This is indeed a very valuable suggestion by the reviewer. We have now used the *Pseudomonas putida* TssA1, whose activity we have already established in previous

publication (Bernal *et al.*, *ISME J*, 2017, 11(4):972-987). We generated chimeras in which the N terminus of *P. aeruginosa* TssA1 was replaced by the cognate domain of *P. putida* TssA1, and reciprocally replaced the C terminus of *P. aeruginosa* TssA1 by the cognate C terminus of *P. putida* TssA1. We have assessed the expression of these chimera using western blotting of HA-tagged version and both are produced although the former far lower than the latter. We then tested the chimera for functionality in the *P. aeruginosa* H1-T6SS, both in competition assay and for Hcp1 secretion. As expected, the TssA1 from *P. putida* (TssA^{PP}) does not complement a *P. aeruginosa* *tssA1* (TssA^{PA}) mutant, however the chimera in which the TssA1^{PP} Nt1 domain has been replaced by that of TssA1^{PA}, then killing efficiency is fully restored for the *P. aeruginosa* *tssA1* mutant. Yet the chimera in which it is the C-terminal domain has been switched, then no complementation of a *P. aeruginosa* *tssA1* mutant is observed. This is in full agreement with other chimera we have engineered in this work, where the C-terminal domain appeared promiscuous and replaceable, but the Nt1 domain provides specificity and cannot be exchanged. All these data are now compiled in Figures 6e and f, with accompanying Supplementary Figure 7.

While obvious to readers familiar with the used technique, it would be good to include some explanation for the colour changes observed in the BTH experiments.

We have now added a sentence in the figure legend and the methods section.

Loop and hairpin comparisons for TssA1 and TssA3 are missing RMSD values.

The comparison of TssA1 and TssA3 Nt1 domain models is now colour coded according to RMSD.

Reviewer #2 (Remarks to the Author):

This study focuses on a description of chimeric TssA proteins produced by Pseudomonas aeruginosa. TssA are gp6-like key oligomeric components of the type 6 secretion system hexagonal inner baseplate, which are involved in sheath polymerization. The main findings are (i) TssA are modular proteins with two domains, (ii) the N terminal domain binds to sheath proteins via two regions, (iii) the C terminal domain control ring formation/oligomerization and can be swapped from TssA subgroups.

The work is incremental or at least confirmatory of a set of previous studies. Some of the data presented here is similar to data published by others.

There is no doubt that TssA is one of the most intriguing proteins within the T6SS. It has a very variable domain organisation, its function has been variably discussed including three rather striking distinctive roles, which could be exclusive or combined within the same protein (Schneider *et al.*, *EMBO J*. 38(18)e100825). As a result, and as pointed by the reviewer in the comments below, there are many questions that still need to be addressed, and answers to these would not be confirmatory or incremental but will advance the T6SS field.

- Figure 3: The interaction of TssA1 with TssBC proteins has been shown in previous studies including one of the authors themselves (Planamente et al 2016). The requirement for domain Nt1 is also documented (Zoued et al. 2016).

Yes, we have reported the interaction between full length TssA1 and TssB1 (Planamente *et al.*, 2016; pull-down and not BTH), and here we used this interaction as a control for testing more systematically the different T6SSs in *P. aeruginosa* and subsequently refine domain

interactions. The reviewer is misled when stating that the requirement for Nt1 is documented. In the publication by Zoued and collaborators. What is shown in this study is an interaction with TssBC involving Nt1-Nt2 domains together. The authors then proposed that the interaction involves Nt2 domain, and this latest assumption is not based on experimental data, but on simulation and docking. Our work comes here with a very different conclusion in establishing that it is Nt1, not Nt2, which provides the interaction specificity with the sheath. This makes sense since short TssAs do not have a Nt2 domain and still interact with the sheath, clearly through their Nt1 domain. All of this is not confirmatory of previous data, but instead brings in a conceptual change.

- Figure 4: Structural information on oligomeric states of TssA subgroups, and the contribution of the C terminal domains in association has been already provided with higher resolution (Dix et al. 2018).

Absolutely, we agree with the reviewer and we have extensively cited this paper. The added value of our data is simply to provide a direct comparison between two TssAs, supposedly of the same group (short TssAs in the same bacterium), but whose behaviour is not strictly identical, again suggesting that the structure and role of TssAs across T6SSs is likely very variable and subtle. This opens the field to consider that the detailed structure of every single TssA might provide valuable information on how it operates and is worth obtaining at high resolution. Assuming that all is wrapped and understood with this one single paper by Dix and collaborators might be an overstatement.

- Figure 9: TssA-sheath molecular docking simulations have been published (Schneider et al 2019) and the author's model of the role of TssA for sheath polymerization has been proposed (Schneider et al 2019, Zoued et al 2016). What is surprising is that the authors model TssA bound to the distal end of the sheath while they showed that is a component of the baseplate, likely to be located at the basal end of the sheath.

As for the published molecular docking simulations, and as addressed in our answer above, it was incorrectly using the Nt2 domain, which is absent in short TssA proteins. The docking may appear similar because there are structural similarities, which we describe in our manuscript, but the fact is that it is the Nt1 domain and not the Nt2 domain that is relevant for the interaction.

Regarding TssA as a member of the baseplate, this is a long debate, but which is not incompatible with TssA moving along with the sheath. In fact, it has been widely described that TssA would be the first component assembled at the membrane complex and recruiting all other components of the baseplate and notably TssK. At this stage there are interactions, and these interactions take place within the baseplate, which is thus in agreement with our previous statement and shown in other publications reporting *P. aeruginosa* TssA1 at the T6SS initiation site (Schneider et al., 2019). What we have missed in our initial publication, is the concept of TssA dynamic and indeed it is now well documented that TssA moves away from the baseplate and stay at the distal end of the sheath. It was then proposed that the difference in behaviour may relate to TssA being a short or long version. However, we have since published (Bernal et al., *PNAS*, 2021), notably using *P. putida* TssA1, that short TssAs also go along with the distal end of the sheath, which was since also shown by the Basler group for the short TssA of *A. baylyi* (Lin et al., *EMBO J.*, 2022). In all, this is again flagging the complexity of TssA structure-function, and should not be seen as conflictual data, but rather as complementary data that should all be taken on board and built on to help the field resolving

the challenge posed by TssA function. We have now made an attempt to summarise this complexity and evolution of our understanding in our introduction so it may be clearer to those who may have missed the stepwise evolution in our understanding of TssA positioning.

Noteworthy the lack of scholarship at several instances in the manuscript is blatant, e.g. “A remarkable feature of TssA proteins is the ability to form multimers, which was PROPOSED to occur through the C terminal domain^{41,42}”. But this was DEMONSTRATED! In Ref 41 the structures of various TssA were reported demonstrating the role of the C terminal domains in oligomerization. And other structures demonstrating this were published in references 19, 21, and 40.

With all due respect to the reviewer the utilisation of “proposed” instead of “demonstrated” was indeed clumsy and not intentionally meaning that it was not experimentally supported. We apologise if this may have been taken this way. In fact, we were actually the first (our EMBO J paper from ref 42), together with the Cascales lab (Zoued et al., 2016, Ref 21), to DEMONSTRATE the multimeric organisation of TssA. We have now changed this in the text, and the reviewer should be reassured that there is no work published on TssA that we are not referring to in our present paper.

The novelty of this work resides and is therefore limited to (i) the observation that TssA binding to the sheath requires the two sheath subunits and two discrete regions of TssA and (ii) to swapping experiments demonstrating the C terminal is functionally permissive.

We appreciate that the reviewer identifies novelties in our work. There is one more point which is not highlighted here, and that we discussed before. This is our demonstration that the specificity goes toward Nt1, not Nt2, which thus reconciles the functional mechanism associated with short TssAs which are devoid of Nt2 domain, and this should be seen as a conceptual change as compared to what was previously published.

Some essential controls are missing, eg the authors use the BTH assay to conclude about the stability of their chimeric constructs, however in the BTH assay the chimeras are fused to stable domain and are produced in E. coli, which does not reflect the in vivo case, in P. aeruginosa. Statistical analyses are missing (Fig 2).

We do not have any valuable specific antibodies against TssAs. So to provide further information on the stability of our chimera, we have now HA-tagged the chimeric proteins and performed western blot analysis. While we were unfortunately unable to tag TssA^{3Nt1}-TssA1^{CTD}, all other chimeric constructs were tagged and their expression assessed in *P. aeruginosa*, including those produced to address other comments. Western blots showing the expression of these chimeric proteins are now shown in Supplementary Figures 6, 7a and 9c.

There is a number of discrepancies with a previous study from the same authors, eg Planamente et al 2016 showed that “TssA1 is not strictly essential for TssB1C1-sheath assembly” while it seems essential in this study.

As addressed in a response to reviewer 1, there is a subtle difference between being essential for a functional T6SS (resulting in effective T6SS secretion) and residual assembly of T6SS sheath. This is what we have published in our previous paper (Planamente et al., 2016), and which is again reported here.

In addition the authors previously showed that Pseudomonas TssA1 has topology/structural similarities with bacteriophage gp6 and is likely to be a structural subunit of the type 6 secretion baseplate interacting with the first layer of sheath (Planamente et al 2016: “The existence in the T6SS of a gp6-like domain that we describe here within TssA1 is a breakthrough since gp6 is central to the phage baseplate organization.”), and this is not discussed in this work. They showed that TssK is an gp8 homologue (Planamente et al 2016) and interactions between TssA domains and the gp8 homologue is not tested here. The gp6 inner baseplate ring is asymmetric (Taylor et al 2016) and is comprised of 12 copies. The comparison between gp6 and TssA oligomeric states and symmetry could be discussed here. This is not clear to me as the authors mention in their 2016 article that “It is thus clear that the TssA protein that we describe in our study, TssA1, and EcTssA (or TssA2) are two distinct proteins and most of the hypothetical mechanism that has been associated with EcTssA remains to be experimentally validated in other T6SSs.” It seems to me that the authors now suggest that their TssA1 protein has a function similar to EcTssA and therefore they validate the EcTssA hypothetical mechanism. This should be discussed in more details and if TssA is not a gp6-like baseplate protein this should be clearly mentioned and corrected in this work.

The reviewer should be commended for such a detailed reading of our previous manuscript (Planamente *et al.*, 2016), but we will say that this should apply to all of the TssAs manuscript that have been published since then, including our manuscript published in PNAS in 2021, which is addressing most of the above comments. We have also addressed, in the above answers, some of the points about comparison between short (*P. aeruginosa* TssA1) and long (EcTssA) TssAs, which we hope rationalise some of the apparent discrepancies. It is important to note that the TssA oligomeric status and symmetry is variable as discussed in the paper by Dix and collaborators (2018), other TssA manuscripts, and also summarised in Ali and Lai (BioEssays, 2022), which we have now added as a reference.

The reviewer is right that we have not generalised the TssA interaction with the gp8 homologue, TssK. We have now performed this experiment using BTH, which confirmed interaction for both TssA2/TssK2 and TssA3/TssK3, and is summarised in Supplementary Figure 3b.

As for the similarity with gp6 there is no reason for this not to stand. We have not stated that TssA is a strict functional gp6 homologue but that TssA has a domain with gp6 similarity, the C terminus, and both are involved in oligomerisation. It is important to note that when comparing phage and T6SS proteins, there are many cases of chimeric situations. For example, gp18 is one protein corresponding to a combination of TssB and TssC, VgrG is a combination of gp5 and gp27 etc..., which means that there is no straightforward and direct one to one functional comparison between these proteins. For example, in our previous paper (Planamente *et al.*, 2016) we have also proposed that gp6 could be a combination of TssA and TssF domains.

Reviewer #3 (Remarks to the Author):

In this paper, Fecht et al compared the TssA proteins among three T6SSs in Pseudomonas aeruginosa. Through genetic mutation and complementation experiments, the authors found that the TssA proteins were specific to their corresponding T6SSs by interacting with cognate TssB/C sheath subunits. By constructing different TssA chimeras, the authors demonstrated that the Nt1 domain of TssA played a pivotal role in this process. Further comparison of the Nt1 domains of TssA1 and TssA3 revealed the presence of specific loop and hairpin secondary structures. Overall, this study provides valuable information for understanding the specific

interactions between the TssA protein and the TssB/C sheath. However, some of the conclusions need more experimental support.

Major issues:

1. Figure 2b: why did the tssA1 mutant exhibit stronger T6SS-mediated killing abilities compared with the tssB1 mutant? Besides, for the complementation experiments, the author should validate the expression of those TssA proteins and the TssA chimeras.

As indicated in previous answers above, the requirement for TssA might not be as stringent as it is for TssB, with some residual activity possibly observed, especially in various genetic background. In this particular case we use a *rsmA/rsmN* background which results in much higher T6SS activity than genetic backgrounds we used in previous papers.

As addressed in previous comments, we have now HA-tagged the chimeric proteins and their production is shown in Supplementary Figures 6, 7a and 9c.

2. For the cryo-EM study of the C-terminal domain of both TssA1 and TssA3, more linearized shapes were observed in TssA3CTD compared to TssA1CTD; the authors mentioned this may be due to flexibility. Did the authors try other truncated versions of TssA3CTD to improve the stability? In the manuscript, the authors claimed that the TssA3CTD presents a 12-fold symmetry arrangement. However, in Fig. 4d and f, the resolution is poor and difficult to conclude.

As for truncating TssA3CTD, we did not try but were quite unsure how we would truncate it, given that it is only be predicted to be 4 helices, in 2 pairs, where both pairs are expected to be essential to make the ring.

We have amended the text to make it clear that we only estimate 12-fold symmetry of TssA3CTD.

3. For the long TssA, what's the role of the Nt2 domain? In the introduction, the authors mentioned the Nt2 domain tends to form dimers surrounding the C-terminal domain ring of TssAL. Does this Nt2 domain participate in the formation of the ring structure? When the authors investigated the structure of TssA2CTD by cryo-EM, did the authors try to include the Nt2 domain in the C-terminal domain for structural study?

The role of Nt2 has not been clearly established, and we show here that it is Nt1 rather than Nt2 that is giving the interaction specificity with the sheath. Previous data have demonstrated that ring formation can take place in the absence of the Nt2 domain, suggesting that it surrounds the core ring, but is not required for ring formation (Dix et al., 2018). Yet, we do suggest in the discussion that it may act as a spacer to ensure proper Nt1 positioning, although we cannot exclude that it may have other roles.

As for the structure of TssA2CTD we also tried to obtain a CTD including the Nt2 domain but as for the CTD alone, the truncated form was well expressed but lost during the purification process, suggesting poor stability.

4. The experimental design must be more convincing to prove that the hairpin and loop regions of Nt1 are responsible for the specificity in sheath interaction. For example, in Fig. 8b, the authors should include the detection of the interaction between TssA1-TssA3hairpin and loop and TssB1/TssC1. To make the conclusion more solid, the authors should perform the interaction assay with TssB3/TssC3. Similarly, the authors should conduct the binding assay of the TssA3-TssA1 variants with TssB1/TssC1 in Fig. 8c.

The hairpin region is found within an extension to the Nt1 domain that is present in short TssA proteins but absent in long TssA proteins. Similarly, bioinformatic analysis carried out previously indicates the region that we know identify as the loop is specific to short TssA proteins (Dix et al., 2018). We therefore propose that these regions may only confer interaction specificity in short TssA proteins. However, at this stage this is an early hypothesis, and we agree additional data will be required to confirm it. We have made it more clear in the manuscript that our data just suggest that these features might be important for sheath interaction.

5. Figure 7: it is interesting that TssA chimeras lacking the Nt2 domain (TssA2Nt1-TssA1CTD and TssA2Nt1-TssA3CTD) were able to partially restore the H2-T6SS activities of tssA2 mutants. This raises the question of whether the Nt2 domain is essential for H2-T6SS activities. It would be interesting to test whether a TssA2 mutant lacking the Nt2 domain can also restore H2-T6SS activities in the tssA2 mutant or whether tssA2 mutants expressing TssA2Nt1-Nt2-TssA1CTD / TssA2Nt1-Nt2-TssA3CTD exhibit higher H2-T6SS activities.

This is a really good point and we have now generated TssA2Nt1+Nt2-TssA1CTD, and TssA2Nt1+Nt2-TssA3CTD chimeras, and tested complementation of *tssA2* mutant using bacterial killing assay which is now shown in the new Supplementary Figure 9. Both HA-tagged chimera are nicely produced as shown by performing western blot analysis. Bacterial killing assay shows that complementation is effective while all chimera are also very active, and there are no significant differences when the N terminal part of the TssA2 chimera carries Nt1 alone or Nt1/Nt2.

6. Figure 8: the authors identified specific loop and hairpin secondary structures within the Nt1 domain of TssA1 and TssA3 that are essential for proper and specific interaction between TssA and the T6SS sheath. However, it remains unclear whether these secondary structures are unique to TssA1 and TssA3 or if they also exist in other TssA homologues and contribute to the interaction with the sheath. Further investigation into the conservation of these structural motifs across different TssA proteins would provide important insights into the mechanism of T6SS assembly.

As stated above, these regions appear to be specific to short TssA proteins based on previous bioinformatic analysis (Dix et al., 2018).

7. Figure 9: the author proposed a docking model of TssA proteins with the sheath using the ZDock server. However, it is unclear why the TssA proteins can only interact with their cognate sheath but not with TssB or TssC independently. Additionally, the key residues or regions of TssB/C that contribute to the interaction with TssA were not identified. To address these gaps, the authors are recommended to test the interacting regions based on the current docking model in order to identify the specific regions of TssB/C that are responsible for the interaction with TssA.

Previously published pull down data has shown that there can be interaction between TssA proteins and sheath components individually. We are not suggesting that interaction only occurs when both sheath components are present, but that it is likely providing the best interface for stable interaction with TssA.

8. The authors should revise Fig. 10 to make it more understandable.

We have edited this figure.

Minor comments:

1. *It is recommended that the authors consecutively number all lines of the manuscript for ease of reference.*

This has now been done.

2. *P. aeruginosa contains three sets of T6SS, including H1-T6SS, H2-T6SS and H3-T6SS. To demonstrate each TssA is specific for its cognate T6SS, Hcp secretion and bacterial competition assays were performed for both TssA1 and TssA2, but the assembly of T6SS sheath structures was applied for TssA3. The authors should explain why these different methods were used.*

We have no secretion readout for the H3-T6SS and that is why we used sheath assembly as an alternative readout for T6SS activity.

3. *There is a spelling mistake in the paper title, and it should be "specificity".*

Very sorry that we have missed that and it has now been rectified.

4. *The format of references is not consistent, particularly for the title of the references. Also, the bacterial names and specific words like "in vivo" should be italic.*

Again, sorry here and that is just the result of the automatic formatting using EndNote. All references have now been reviewed and made consistent.

Reviewer #3 (Remarks to the Author):

The authors have fully addressed my concerns.